# Effects of treated wastewater on the ecotoxicity of small streams – Unravelling the contribution of chemicals causing effects

Cornelia Kienle[1]*, Etiënne L. M. Vermeirssen[1], Andrea Schifferli[1], Heinz Singer[2], Christian Stamm[2], Inge Werner[1]

1 Swiss Centre for Applied Ecotoxicology, Dübendorf, Zürich, Switzerland, 2 Department of Environmental Chemistry, Swiss Federal Institute of Aquatic Science and Technology (Eawag), Dübendorf, Zürich, Switzerland

* cornelia.kienle@oekotoxzentrum.ch

**Data Availability Statement:** The data underlying the results presented in the study are available

## Abstract

Wastewater treatment plant effluents are important point sources of micropollutants. To assess how the discharge of treated wastewater affects the ecotoxicity of small to medium-sized streams we collected water samples up- and downstream of 24 wastewater treatment plants across the Swiss Plateau and the Jura regions of Switzerland. We investigated estrogenicity, inhibition of algal photosynthetic activity (photosystem II, PSII) and growth, and acetylcholinesterase (AChE) inhibition. At four sites, we measured feeding activity of amphipods (*Gammarus fossarum*) *in situ* as well as water flea (*Ceriodaphnia dubia*) reproduction in water samples. Ecotoxicological endpoints were compared with results from analyses of general water quality parameters as well as a target screening of a wide range of organic micropollutants with a focus on pesticides and pharmaceuticals using liquid chromatography high-resolution tandem mass spectrometry. Measured ecotoxicological effects in stream water varied substantially among sites: 17β-estradiol equivalent concentrations ($EEQ_{bio}$, indicating the degree of estrogenicity) were relatively low and ranged from 0.04 to 0.85 ng/L, never exceeding a proposed effect-based trigger (EBT) value of 0.88 ng/L. Diuron equivalent ($DEQ_{bio}$) concentrations (indicating the degree of photosystem II inhibition in algae) ranged from 2.4 to 1576 ng/L and exceeded the EBT value (70 ng/L) in one third of the rivers studied, sometimes even upstream of the WWTP. Parathion equivalent ($PtEQ_{bio}$) concentrations (indicating the degree of AChE inhibition) reached relatively high values (37 to 1278 ng/L) mostly exceeding the corresponding EBT (196 ng/L $PtEQ_{bio}$). Decreased feeding activity by amphipods or decreased water flea reproduction downstream compared to the upstream site was observed at one of four investigated sites only. Results of the combined algae assay (PSII inhibition) correlated best with results of chemical analysis for PSII inhibiting herbicides. Estrogenicity was partly and AChE inhibition strongly underestimated based on measured steroidal estrogens respectively organophosphate and carbamate insecticides. An impact of dissolved organic carbon on results of the AChE inhibition assay was obvious. For this assay more work is required to further explore the missing correlation of bioassay data with chemical analytical data. Overall, the discharge of WWTP effluent led to

from: https://zenodo.org/record/3546324#. XeGi6m5FwuU or DOI: 10.5281/zenodo.3546324.

**Funding:** Authors received funding from the Swiss Federal Office for the Environment (https://www. bafu.admin.ch), the Eawag (https://www.eawag.ch) and the Swiss Centre for Applied Ecotoxicology (https://www.ecotoxcentre.ch).

**Competing interests:** The authors have declared that no competing interests exist.

increased estrogenicity, PSII and AChE inhibition downstream, irrespective of upstream land use.

## Introduction

Micropollutants are organic and inorganic substances, which occur in very low concentrations in surface waters (ng - µg/L range). Even at these low concentrations, some of them can elicit effects on aquatic organisms [1]. Herbicides, such as diuron or terbutryn, and other plant protection products, can inhibit the growth and photosynthesis of algae and aquatic plants [2, 3]. Neurotoxic chemicals such as organophosphate and carbamate insecticides or neonicotinoids may lead to enzyme inhibition with subsequent behavioural effects on aquatic invertebrates as well as vertebrates [4–8]. Pharmaceuticals, such as endocrine active substances, antibiotics or anti-inflammatory drugs, were shown to affect various organs, reproduction and growth of aquatic vertebrates (e.g. [9, 10]).

Wastewater treatment plant (WWTP) effluents are important point sources of micropollutants, and the input of micropollutants from WWTPs leads to a frequent exceedance of effect-based ecotoxicological environmental quality standards (EQS) for single chemicals in the receiving waters (e.g. [11–14]). Additionally, due to the presence of a complex mixture of micropollutants in stream ecosystems, aquatic organisms are experiencing an increased risk of mixture toxicity [11]. To date, numerous studies have been performed assessing the ecotoxicological risks of single groups of micropollutants (e.g. estrogenic substances [9], herbicides [15] or neurotoxic chemicals [16, 17]) in the laboratory as well as in the field. In addition, effects of complex mixtures of micropollutants have been investigated in the laboratory or in *in situ* flow through studies [2, 18, 19]. These studies showed that micropollutants released into streams from WWTPs pose a potential threat to biota in aquatic ecosystems (e.g. [20–24]).

While the impact of micropollutants on a small number of species is relatively well documented, studies assessing ecotoxicological risks of micropollutants directly in the ecosystem are scarce. Pesticides led to changes in the taxonomic composition of macroinvertebrate communities (e.g. [25–27]) causing a loss of sensitive species and an impairment of leaf litter degradation (e.g. [28]). In the projects "Schussenaktiv" and "Schussenaktivplus" [21], *in vitro* as well as *in vivo* bioassays were shown to be suitable for the assessment of micropollutant effects on resident organisms. Amphipods displayed a reduced fecundity in a wastewater-impacted stream [29] In addition, deleterious effects on reproduction and endocrine disruption in snails and fish reflected the potential for endocrine disruption measured by *in vitro* bioassays in stream water samples [30]. Similar results have been observed with regard to genotoxic, dioxin-like and embryotoxic effects measured in environmental samples the laboratory which reflected effects in wild fish [31]. These results show the potential of ecotoxicological bioassays conducted in the laboratory to predict effects in the field. To date, such studies were mostly performed assessing single streams. However, as composition and concentrations of the mixture of micropollutants may vary considerably among WWTPs, studies assessing micropollutant concentrations and effects over a number of streams are needed. In addition, the measured chemicals should be linked to the observed effects. The occurrence of a complex mixture of potentially thousands of micropollutants in stream ecosystems (e.g. [13, 32]) makes it difficult to quantify the level of pollution.

The present study was performed as part of the project EcoImpact which aims to address the effects of multiple stressors on stream ecosystems by using semi-field experiments and

field investigations [33]. The large scale field study, which was conducted at 24 wastewater treatment plants and adjacent rivers across the Swiss Plateau and the Jura in 2013 and 2014, allows to draw conclusions on concentrations and effects of micropollutants in stream ecosystems up- and downstream of wastewater treatment plants for a high number of sampling sites with diverse contexts. These are: the combination of exposure and effect assessment, the possibility to detect general trends, and the ability to quantify variability across sites. The results will be used to configure a monitoring scheme for Swiss rivers prior to upcoming nationwide WWTP upgrade to ozonation or activated carbon treatment (or a shutdown of the plant) [33].

Munz, Burdon [34] performed an environmental risk assessment on the acute risk of all chemicals measured at the 24 sites and evaluated the potential impact of WWTPs on receiving ecosystems (acute toxic pressure). These results were then validated with macroinvertebrate biomonitoring data. The authors found that pharmaceuticals were the dominant micropollutants downstream of the WWTPs, however the acute toxic pressure was mainly due to pesticides. Overall, much of the total ecotoxicological risk was, in general, caused by five single compounds with diclofenac, diazinon, and clothianidin being the most relevant. These findings were positively correlated with aquatic macroinvertebrate sensitivity to pesticides. Burdon, Munz [35]] found that water quality and modified habitat explained 30 and 13% of the composition of the macroinvertebrate community, respectively. Pesticides in particular explained 3% of this community composition and, except for oligochaetes, agricultural land-uses (e.g. arable cropping) had a stronger impact on the organisms than wastewater. In gammarids sampled at 10 of the 24 sites internal concentrations of organic micropollutants were increased downstream of the WWTP compared to the corresponding upstream site [36]. At three of the 24 sites, Neale, Munz [37] assessed ecotoxicological effects with a variety of bioassays. They showed that the extent to which effects in bioassays can be explained by chemical analysis depends on the respective effects / compound groups. For example, for PSII inhibiting herbicides the majority of the effects was explained by the compounds measured. For other bioassays (e.g. measuring oxidative stress response or an activation of the androgen receptor) little of the observed effects were attributable to the chemicals measured.

The present study complements previous studies by comparing selected ecotoxicological effects (estrogenicity, inhibition of photosynthetic activity and neurotoxicity due to impaired acetylcholinesterase) with chemical analysis at all 24 EcoImpact sites. We investigated whether effects differed upstream and downstream, how well the results of the bioassays corresponded to results of chemical analysis and evaluated what proportion of ecotoxicological effects measured across a large number of sites can be explained by chemical analysis. Based on the bioassay results an environmental risk assessment was performed and compared to risk assessment results based on chemical analytical data. This presents a re-evaluation of data from Munz, Burdon [34] with a focus on compounds eliciting effects in the applied *in vitro* bioassays. In addition, at four sites *in situ* feeding activity of amphipods (*Gammarus fossarum*) was investigated and reproduction of water flea (*Ceriodaphnia dubia*) was assessed *in vivo* using samples taken up- and downstream of WWTP effluent discharges.

## Materials and methods

### Site selection and sampling locations

Twenty-four medium sized streams affected by WWTP inputs were selected as described in [33, 35]. In brief, selected sites had no WWTP effluent upstream (only site #22 (Val-de-Ruz) had a small fraction of wastewater upstream as we detected later on), less than 21% of urban areas and less than 10% of specialty crops (e.g. fruits) upstream of the WWTP, as well as more than 20% effluent downstream during low flow conditions (at $Q_{347}$; this represents the

discharge which, averaged over 10 years, is reached or exceeded on average at 347 days per year and which is not significantly affected by water congestion, abstraction or supply; 95% low-flow conditions [38]). In addition we aimed at: similar stream morphology, riparian land use and vegetation at up- and downstream reaches. Catchments were distributed across three Swiss biogeographical regions (Swiss Plateau, Jura, and Pre-alps) and differed considerably in land-use composition. All WWTPs were operated using activated sludge treatments as secondary treatment with one exception (Colombier, using a trickling filter system) [34].

At each study site, one upstream (US) location was chosen as reference site and one downstream (DS) sampling location as impacted site. Additionally, the WWTP effluent was sampled. The upstream site was located some 20 m upstream from the effluent discharge. At the downstream sampling site wastewater and river water were completely mixed. Further details on the sampling locations can be found in the SI (Table A and Figs A and B in S1 Appendix) and in [33, 34, 39]. All water bodies and the riparian areas are public ground. We asked all cantonal authorities involved and all WWTP operators for their consent. Given established working relationships between Eawag and these institutions, the contacts were mostly by informal phone calls. After finishing the water quality measurements, all WWTP operators and the respective cantons received a report with the final data and a short interpretation of the results.

## Sampling

For chemical and ecotoxicological analyses grab samples were taken at low-flow (dry weather) conditions in June 2013 for the first half of sites and in May 2014 for the second half of sites as described in [34]. For chemical analysis (except for estrogenic compounds) one additional sampling of the first half of sites took place beginning of 2014 and five additional samplings of the second half of sites in 2014 and beginning of 2015. These additional data were included in the current study to evaluate the occurrence of compounds which are relevant in the applied bioassays, as well as to perform an environmental risk assessment based on chemical analysis results. Further details on sampling dates and corresponding analyses are provided in Table B in S1 Appendix.

Bottles for bioassays and chemical analysis were rinsed with acetone (p.a.) or methanol (p. a.) prior to use and left to evaporate to dryness. All other material for sampling was rinsed three times with acetone (p.a.) before use. Before filling, all bottles were rinsed three times with water from the sampling site. Grab samples in the river were taken from different points across the width of the river using a 1 L PTFE scoop. Five litres of sample were collected in a 5 L glass sampling bottle, mixed thoroughly and subsequently distributed in 0.5 or 1 L portions to 1 or 2 L glass bottles for chemical analysis and bioassays, respectively.

Samples were transported to the laboratory at 5–8°C in cooling boxes filled with ice. Samples were either stored at 2–8°C until further analysis within 24 h (for general water chemistry), or frozen at -20°C on the day of sampling and stored until further processing. Storage of samples did not exceed 3 months.

## Chemical analyses

**General water chemistry.** Several general water chemistry parameters were measured within 24 h after sampling and are reported in [34, 35, 39]. Analysed parameters included conductivity, pH, alkalinity and hardness as well as the determination of nutrients and major ions. Analyses were performed using standard methods described for the Swiss National River Monitoring and Survey Programme. Based on all measured water quality parameters dilution coefficients for each sampling site were calculated (see Table C in S1 Appendix).

**Chemical analysis of micropollutants.** Methods and results have been published by Munz, Burdon [34]. A complete list of measured micropollutants, corresponding limits of quantification (LOQ) and concentrations, as well as further details on the analysis procedure can be found there. The applied methods are briefly recapitulated here. In 2013, WWTP effluent samples were analysed for 389 organic micropollutants (including transformation products). Analysed substances included 188 pharmaceuticals, 143 plant protection products, 19 biocides, 15 anaesthetics, 4 industrial chemicals, 4 corrosion inhibitors, 2 personal care products, and 1 tracer. In the 2014 assessment programme a priority mixture of 57 compounds based on their relevance for Swiss streams was analysed including 32 plant protection products (incl. 1 transformation product), 21 pharmaceuticals, 2 corrosion inhibitors (incl. 1 transformation product), 1 biocide (personal care product), 1 food additive, and caffeine, a tracer for untreated effluent [34].

Frozen samples were left to thaw in the dark overnight, subsequently filtered with glass fibre filters by using a vacuum filtration unit and acidified to pH 3 using 0.1% hydrochloric acid (HCl, p.a.). Samples were either enriched with offline SPE as described in [40] using manually packed mixed-modes cartridges [41] (June 2013 samples) or using an automated online SPE as described in [42] (May 2014 samples). This was followed by liquid chromatography high-resolution mass spectrometry (LC-HRMS) and subsequent quantification according to [41, 42].

**Estrogens in WWTP effluent.** Five estrogenic substances were analysed in WWTP effluent samples including the natural estrogens estrone and 17β-estradiol, the synthetic estrogen 17α-ethinylestradiol, as well as the industrial chemicals bisphenol A and nonylphenol. The selection of estrogens was based on previous studies, where estrone, 17β-estradiol, 17α-ethinylestradiol, alkyphenols (e.g. nonylphenol), and bisphenol A were identified as the most relevant estrogenic substances present in WWTP effluents [43–45].

Frozen samples were left to thaw in the dark overnight, subsequently filtered with glass fibre filters by using a vacuum filtration unit and acidified to pH 3 using 0.1% hydrochloric acid (HCl, p.a.). Samples were enriched as described in [18]. Five hundred mL of effluent sample was concentrated 2500 times using LiChrolut® EN-RP18 cartridges (Merck, Germany), purified with mini silica gel columns and stored in 200 μL ethanol at -20°C until use. Details on the methods applied are provided in Table D in S1 Appendix. Analyses of samples was performed by liquid chromatography coupled with mass spectrometry (LC–MS/MS) according to [46] (API 4000 LC–MS/MS, Applied Biosystems, USA). Overall, this method was less robust than the method used to determine the other compounds. An important reason is that most of the data had to be extrapolated between 0 and the lowest calibration standard. Further information on the applied method is provided in Table E in S1 Appendix.

## Ecotoxicological bioassays

Ecotoxicological effects were investigated in the laboratory as well as in the field. At all sites, estrogenic activity (Yeast Estrogen Screen / ERα-CALUX®), effects on the photosynthesis and growth of single-celled green algae (*Raphidocelis subcapitata*) (combined algae assay), and neurotoxic effects (AChE inhibition assay) were evaluated. At four selected sites, feeding activity of amphipods (*Gammarus fossarum*) *in situ* as well as water flea (*Ceriodaphnia dubia*) reproduction *in vivo* in water samples were assessed (see further information in sections S4.1 and S4.2 in S1 Appendix).

**Sample preparation.** Samples were left to thaw in the dark overnight, subsequently filtered with glass fibre filters by using a vacuum filtration unit and acidified to a pH of 3 using 16% hydrochloric acid (HCl, p.a.). Subsequently, samples were enriched as described in [18,

47]. In brief, 500 mL (effluent sample) or 1000 mL (river sample) was enriched 500 and 1000 times respectively using LiChrolut® EN-RP18 cartridges (Merck, Germany), and subsequently stored in 1 mL of a solvent mixture (~50% ethanol, ~50% acetone and methanol) at -20˚C until analysis. Details on the applied methods are provided in Table D in S1 Appendix. Further information on its robustness and limitations is provided in [48].

**Yeast Estrogen Screen.** The Yeast Estrogen Screen (YES) with the recombinant yeast *Saccharomyces cerevisiae* was performed according to [49, 50] in 96 well microtitre plates using yeast cells provided by John Sumpter (Brunel University, Uxbridge, UK). In brief, yeast cells were cultured in growth medium on an orbital shaker at 30˚C for 24 h before the onset of the test. On the test day, the reference substance (17β-estradiol, a very potent estrogen, in ethanol), the water sample extracts, and the solvent control (ethanol: 80 μL/well, n = 8 wells/plate) were pipetted on the plates. Both, the reference substance and the water sample extracts, were tested in triplicates in a 1:2 dilution series with the initial concentration of 17β-estradiol being 1.25 x $10^{-9}$ M and maximum enrichment factors of the samples of 200 (WWTP effluent) and 400 (river water). The solvent was evaporated completely in a sterile bench. In the meantime the cell density of the yeast cells was determined, assay medium prepared and seeded with 4 x $10^7$ yeast cells. Subsequently the yeast cells suspension was pipetted on the test plate (200 μL/well). After 72 h of incubation at 30˚C, cell density ($OD_{620\ nm}$) and colour change ($OD_{540\ nm}$) were measured using a plate reader (Synergy 4, Biotek, Winooski, United States).

In addition to the YES, another *in vitro* reporter gene assay to assess estrogenic activity was performed for the 2014 samples, the ERα-CALUX. It is described in section S4.3 in S1 Appendix.

**Combined algae assay.** The combined algae assay was conducted as described in [18, 49]. In brief, algae cells were cultured in Talaquil growth medium for at least two times 72 h before the onset of the test. On the test day, the reference substance (diuron, a very potent PSII-inhibiting herbicide, in ethanol), the water sample extracts and the solvent control (ethanol: 80 μL/well, n = 8 wells/plate) were pipetted on the plates. The reference substance and water sample extracts were tested in triplicates in a 1:2 dilution series, with the initial concentration of diuron being 3.0 x $10^{-7}$ M and maximum enrichment factors of the samples of 133 (WWTP effluent) and 267 (river water). After a complete ablation of the solvent, the substances were re-suspended in 150 μL Talaquil assay medium. Finally 150 μL of algae suspension with an $OD_{685\ nm}$ of 0.1 were added to each well. Photosynthesis inhibition by means of effective quantum yield was measured using a Maxi-Imaging PAM (pulse amplitude modulation, IPAM) fluorimeter (Walz, Effeltrich, Germany) as described in [49, 51] after 2 h. Growth of algae was measured by means of absorbance at 685 nm in a microtitre plate photometer (Synergy 4) at test start and test end and at two time points in between.

### Acetylcholinesterase inhibition assay

The AChE inhibition assay was performed according to Escher, Bramaz [49]; being based on the method of Ellman, Courtney [52] it was adapted to a 96-well microtiter plate format based upon the DIN standard 38415–1 [53] and Hamers, Molin [54]. In the assay, the production of thiocholine, which is produced as the substrate acetylthiocholine is hydrolysed, is measured colorimetrically. Purified eel acetylcholine esterase (Sigma) was used as the enzyme. First, water sample extracts (in triplicates in a 1:2 dilution series) were pipetted on the plates. Maximum enrichment factors of the samples on the plate were 166 (WWTP effluent) and 333 (river water). After complete ablation of the solvent, the samples were re-suspended in 50 μL phosphate buffer (0.05 M) and the reference substances (paraoxon-ethyl (97.5%, Sigma) and ethyl-parathion (99.7%, Sigma) in phosphate buffer) were added to the plates in unicates in a 1:2

dilution series. The maximum concentration of paraoxon-ethyl and ethylparathion on the plate were $3.0 \times 10^{-6}$ and $2.1 \times 10^{-6}$ M respectively. Subsequently, all wells were oxidised by addition of 5 μL N-bromo-succinimide and supplemented with 5 μL ascorbic acid. To initiate the inhibition reaction 100 μL phosphate buffer and 40 μL AChE solution were added to every well followed by thorough mixing on a plate shaker (MS 3 digital, IKA, Wilmington, USA) for 10 min. After this time 40 μL acetylthiocoline / dithiotonitrobenzoic acid solution were added to each well, and the enzymatic reaction measured at an $OD_{420 \, nm}$ in a plate reader (Synergy 4) for 10 min in 30 s intervals.

## Data analysis

**Calculation of bioanalytical equivalent concentrations.** For the *in vitro* bioassays, bioanalytical equivalent ($BEQ_{bio}$) concentrations were determined, representing the concentration of the reference substance which elicits the same effect as the environmental sample [47, 55]. Naming of $BEQ_{bio}$ varied depending on the reference compound: We determined 17β-estradiol equivalents ($EEQ_{bio}$) for estrogenic activity, diuron equivalents ($DEQ_{bio}$) for algae PSII and growth inhibition, and parathion equivalents ($PtEQ_{bio}$) for AChE inhibition. Limits of detection (LOD) and limits of quantification (LOQ) were calculated as 3-fold and 10-fold the standard deviation (SD) of the averaged induction (YES, ERα-CALUX) or inhibition (combined algae assay, AChE inhibition assay) of the solvent control for each assay plate.

$BEQ_{bio}$ values were calculated for each sample from the effective concentration ($EC_{50}$) or the 10% effect level ($PC_{10}$) of the reference (ref) divided by the $EC_{50}$ of the sample (in relative enrichment factors (REF)) or its $REF_{10}$ as described in [37] and [56] (Eq 1).

$$BEQ_{bio} = \frac{EC_{50}(ref)}{EC_{50}(sample)}(ng/L) \, or \, \frac{PC_{10}(ref)}{REF_{10}(sample)} ng/L \tag{1}$$

The REF is defined as (Eq 2)

$$REF = Concentration \, Factor_{SPE} \times Dilution \, Factor_{bioassay} \tag{2}$$

Data analysis for YES and ERα-CALUX was performed as described in [56]. In general, data with a CV of triplicates ≤ 20% were accepted. Data were evaluated by fitting a concentration response curve using the 4-parameter Hill function (see section S5, Eq 1 in S1 Appendix; GraphPad Prism®, version 5.02 for Windows, GraphPad Software, La Jolla, USA) with $R^2 \geq$ 0.98 as acceptance criterium for the fit. Subsequently, induction data of reference and test sample were normalised (see section S5, Eq 2 in S1 Appendix) and dose response curves of the normalised data fitted from 0 to 100% (with 0% referring to the response in the solvent control and 100% being the response maximum fitted for the reference 17β-estradiol). To determine the concentration of the positive control (PC) needed for 10% effect, this level was interpolated from the normalised reference dose-response curve. For determining the REF (see Eq 2) necessary to produce 10% effect, the 10% effect level was interpolated from the normalised dose response curve of the sample. Finally, the $PC_{10}$ was divided by the $REF_{10}$ to determine the estrogenic activity of the sample (17β-estradiol equivalents, $EEQ_{sample}$) (see section S5, Eq 3 in S1 Appendix), and $EEQ_{bio}$ concentrations reported as ng/L.

Data analysis for the combined algae assay and the AChE inhibition assay was performed as described in [47] and [55] by fitting dose-response curves of the reference substance and the samples using a sigmoidal fit with the slope adjusted to the one of the reference substance (GraphPad Prism®). $BEQ_{bio}$ values were calculated according to Eq 1. $DEQ_{bio}$ and $PtEQ_{bio}$ concentrations were reported as ng/L.

**Comparison of bioassay and chemical data.** To establish what percentage of the effect can be explained by the measured chemicals, mixture toxicity evaluation was performed as described in [37]. In addition, with this evaluation we showed what percentage of the effect can be explained by individual detected chemicals or groups of compounds with the same mode of action.

To enable a comparison of the results from *in vitro* bioassays with the results of the chemical analysis, $BEQ_{bio}$ was compared to $BEQ_{chem}$. $BEQ_{chem}$ was calculated based on the sum of the concentrations of individual compounds ($c_i$) multiplied by their respective compound-specific relative effect potencies ($REP_i$).

$REP_i$ were calculated as follows (Eq 3):

$$REP_i = \frac{EC_{50}(ref)}{EC_{50}(i)} \tag{3}$$

$BEQ_{chem}$ was calculated as follows using the measured chemical concentration ($c_i$) (in ng/L) and the calculated $REP_i$ (Eq 4):

$$BEQ_{chem} = \sum_i^n REP_i \cdot c_i \tag{4}$$

REP for measured estrogens in the YES and the ERα-CALUX, for measured PSII inhibitors in the combined algae assay and for measured acetylcholinesterase inhibitors in the AChE inhibition assay are listed in the SI (section S6, and Tables F, G, and H in S1 Appendix). Relative potencies compared to parathion for carbamate and organophosphate insecticides were determined for 12 of the 14 measured compounds. The relative potencies of dimethoate and fenoxycarb could not be obtained due to solubility problems.

**Assessing the fraction of wastewater effluent downstream with bioassays.** The fraction of wastewater effluent downstream ($f_{eff}$) was determined to compare the measured chemical ($c_i$) or $BEQ_{bio}$ concentrations at the downstream sites with the concentrations expected based on pure physical mixing of upstream water and WWTP effluent as described in [37] according to Eq 5.

$$f_{eff} = \frac{C_{i,\ downstream} - C_{i,\ upstream}}{C_{i,\ effluent} - C_{i,\ upstream}} \text{ or } f_{eff} = \frac{BEQ_{bio,\ downstream} - BEQ_{bio,\ upstream}}{BEQ_{bio,\ effluent} - BEQ_{bio,\ upstream}} \tag{5}$$

To calculate the fraction of wastewater based on chemical analysis, all data for general water chemistry parameters were used, as full data sets of these parameters for both sampling events were available (see Table B in S1 Appendix). $F_{eff}$ was first calculated for each parameter individually, then, the median of these individual values was used as final $f_{eff}$. Table C in S1 Appendix lists the resulting dilution coefficients.

**Risk assessment of chemical analysis and bioassay results.** Risk assessment for chemical analysis was, on the one hand, performed based on the measurements of individual chemicals by comparing their concentrations to chronic annual average environmental quality standard (AA-EQS) (Eq 6) (for an overview on applied AA-EQS values see Tables I, J, and K in S1 Appendix), and, on the other hand, based on a mixture risk quotient ($RQ_{mix}$) for relevant substances (Eq 7) [11, 57]. Estrogenic compounds were measured once at all sites, PSII inhibitors and AChE inhibitors were measured 2 times at the 2013 sites and 6 times at the 2014 sites (see

Table B in S1 Appendix).

$$Risk\ quotient\ (RQ) = \frac{MEC}{AA - EQS} \qquad (6)$$

$$RQ_{mix} = \sum \frac{MEC_i}{AA - EQS_i} \qquad (7)$$

If RQ < 1 → quality criterion kept
If RQ > 1 → quality criterion exceeded
With
MEC = measured environmental concentration
AA-EQS = annual average environmental quality standard

Risk assessment for bioassays was performed according to [58] (estrogenic activity, algae PSII and growth inhibition) by comparing $BEQ_{bio}$ values to effect-based trigger (EBT) values, taking into account available AA-EQS for compounds relevant to the assay as well as their respective REPs and according to [59] (AChE inhibition) by comparing $BEQ_{bio}$ values to AA-EQS of the reference compounds for the respective bioassay (Eq 8).

$$Risk\ quotient\ (RQ) = \frac{BEQ_{bio}}{EBT} or \frac{BEQ_{bio}}{AA - EQS} \qquad (8)$$

If RQ < 1 → quality criterion kept
If RQ > 1 → quality criterion exceeded
With
$BEQ_{bio}$ = Bioanalytical equivalent concentration from *in vitro* bioassay
EBT = effect-based trigger value
AA-EQS = annual average environmental quality standard
Applied EBT values were:

| | |
|---|---|
| Estrogenic effects: | 0.88 ng/L $EEQ_{bio}$ (YES) / 0.1 ng/L $EEQ_{bio}$ (ERα-CALUX®) |
| PSII-inhibiting effects: | 70 ng/L $DEQ_{bio}$ |
| Growth inhibiting effects: | 130 ng $DEQ_{bio}$/L |
| AChE-Inhibition: | 196 ng/L PtEQ (based on AA-EQS of diazinon (12 ng/L) ~16 times more potent than Parathion) (see Table H in S1 Appendix). |

## Results

### Micropollutant target screening and chemical-based risk assessment

In the following, the results for compound groups detected by the bioassays applied will be described, details for a part of these and further compounds are provided in [34]. In addition to PSII and AChE inhibiting compounds (see [34]), results of measurements for estrogenic compounds are reported. This serves as a basis to compare the results of the bioassays with the results of the chemical analysis. In addition, risk assessment for chemical analysis based on the measurement of individual chemicals as well as based on the $RQ_{mix}$ is shown.

**Estrogenic compounds.** This compound group was measured in WWTP effluent only. Four of five estrogenic substances analysed were detected: estrone, 17β-estradiol, bisphenol A and nonylphenol with highest concentrations measured for the last two (Fig 1 and Table L in S1 Appendix). Compared to the natural estrogens, estrone and 17β-estradiol, bisphenol A and nonylphenol have low estrogenic potency *in vitro* (Table F in S1 Appendix) as well as *in vivo* [56, 60, 61].

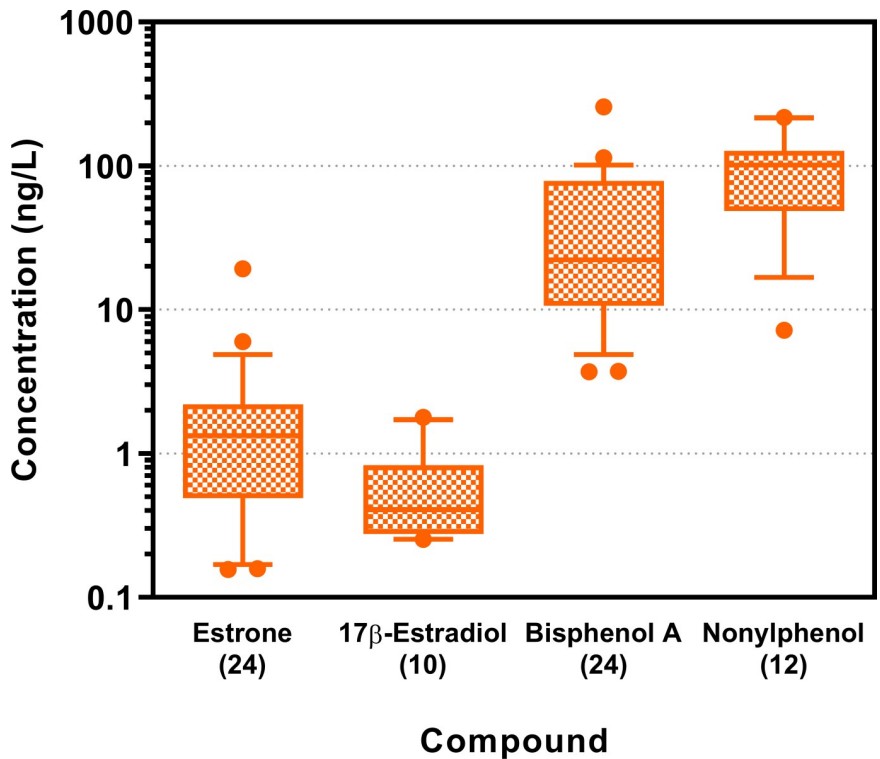

**Fig 1. Concentrations of estrogenic compounds (ng/L) in wastewater treatment plant effluent of all 24 sites assessed in 2013 and 2014.** Box-Whisker plots with the line representing the median, the box the mean 50% of the data and the Whiskers the 10–90 percentile. Dots represent values outside this range. The respective n is given in brackets after the compound name. Displayed are the four compounds measured above the respective limit of quantification (LOQ). All values for 17α-ethinylestradiol were below the LOQ. Due to analytical difficulties no values for nonylphenol could be obtained in 2013. LOQs were 0.1 ng/L (estrone), 0.2 ng/L (17β-estradiol), 0.3 ng/L (17α-ethinylestradiol), 1.6 ng/L (bisphenol A) and 1.2 ng/L (nonylphenol). Data are provided in S1_Data.

As these compounds were only measured in WWTP effluent, the exceedance of AA-EQS for all measured compounds was assessed by extrapolating the concentrations measured in the WWTP effluent to the expected concentrations in the river by applying the respective dilution factor (Table C in S1 Appendix). This was possible for 21 of the 24 sites, due to limited data from the upstream site. 17α-ethinylestradiol was never detected above its LOQ (0.3 ng/L), therefore exceedances of its AA-EQS (0.037 ng/L) could not be assessed. 17β-estradiol concentrations were above the LOQ (0.2 ng/L) in only 10 of 24 wastewater samples, estrone and bisphenol A were quantified at all 24 sites.

Based on extrapolated river concentrations at downstream sampling sites, no exceedances of the quality standard were expected for bisphenol A (AA-EQS: 240 ng/L), whereas the AA-EQS for estrone (3.6 ng/L) and 17β-estradiol (0.4 ng/L) might have been exceeded at one of 24 sites, respectively, and the AA-EQS for nonylphenol (43 ng/L) at two of 12 sites measured in 2014. Due to analytical problems nonylphenol concentrations were not measured in 2013 samples. The combined risk of all estrogenic compounds quantified ($RQ_{mix}$) could have been higher than one at 6 of 24 sites.

**PSII inhibiting herbicides.** Eighteen of the 78 herbicides measured were PSII inhibitors. All of them were detected in the water samples at least once. Compounds occurring most frequently were atrazine, diuron, isoproturon, simazine, terbutryn, terbuthylazine, as well as its metabolite terbutylazine-2-hydroxy, and metribuzin (Fig 2 and Table M in S1 Appendix).

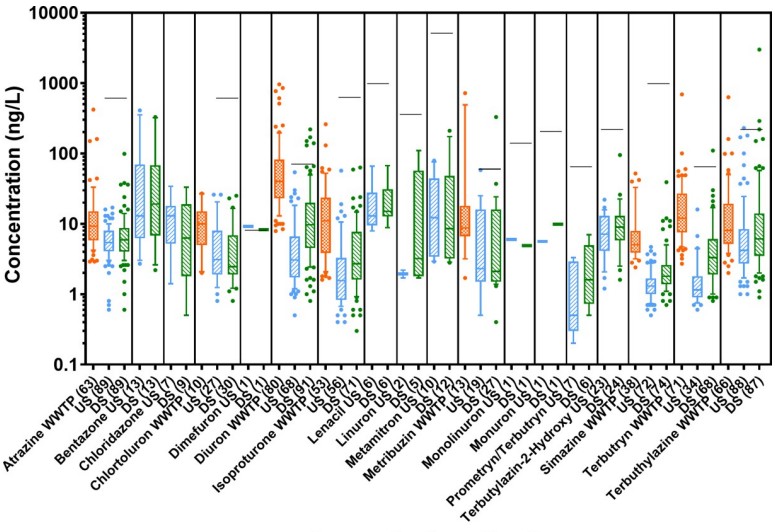

**Fig 2. Concentrations of photosystem II inhibitors (ng/L) in wastewater treatment plant (WWTP) effluent, and at upstream (US) and downstream (DS) locations of all 24 sites assessed in 2013 and 2014.** Box-Whisker plots with the line representing the median, the box the mean 50% of the data and the Whiskers the 10–90 percentile. Dots represent values outside this range. The black line indicates the annual average environmental quality standard (AA-EQS) value for the respective compound (if it was below 10'000). The respective n is given in brackets after the compound name. Limits of quantification were determined individually for each sample and data are reported in detail in [34].

Exceedances of AA-EQS in river samples occurred for 5 PSII inhibitors, namely diuron (5x downstream), dimefuron (1x upstream, 1x downstream), metribuzin (1x downstream), terbutryn (1x downstream) and terbuthylazine (1x upstream, 2x downstream) (see black lines in Fig 2). The RQ$_{mix}$ of one (taking into account all PSII inhibiting compounds quantified) was exceeded 4 times at upstream sites and 15 times at downstream sites (see Fig R in S1 Appendix).

**AChE inhibiting insecticides.** Fourteen of 37 insecticides measured were AChE inhibitors. Eight were detected in the water samples, namely the organophosphates chlorpyrifos, chlorpyrifos-methyl, diazinon, dimethoate, and the carbamates carbofuran, fenoxycarb, methiocarb and pirimicarb. Diazinon was, by far, the compound detected most frequently followed by dimethoate and pirimicarb (Fig 3 and Table N in S1 Appendix). All other compounds were detected in individual samples only.

Six of the 14 measured AChE inhibitors exceeded their respective AA-EQS in river samples (Fig 3). Most of the exceedances were observed for diazinon (1x upstream, 11x downstream). Chlorpyrifos (1x upstream, 1x downstream), chlorpyrifos-methyl (1x downstream), dimethoate (1x downstream), and fenoxycarb (2x downstream) exceeded their respective AA-EQS occasionally. The RQ$_{mix}$ of one was exceeded at two upstream and 13 downstream sites (see Fig S in S1 Appendix).

## Bioanalysis

**Estrogenic activity.** The average estrogenic activity measured in WWTP effluent was 0.83 ng/L EEQ$_{bio}$ (YES). In the river, estrogenicity was lowest at upstream sites (mean: 0.08 ng/L EEQ$_{bio}$), and significantly higher at sites downstream of the WWTP (mean: 0.22 ng/L EEQ$_{bio}$) (Fig 4 and Table O in S1 Appendix) indicating a considerable impact of the WWTP effluent. Results obtained for estrogenic activity using ERα-CALUX are provided in S1 Appendix

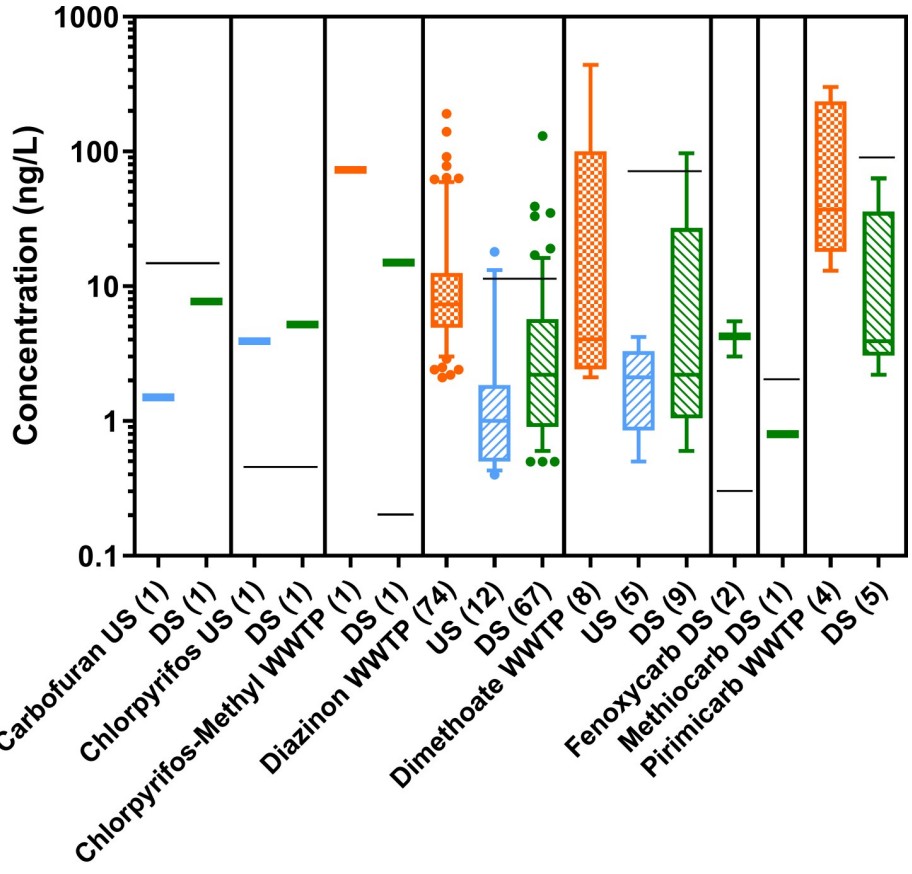

**Compound and sampling site**

**Fig 3. Concentrations of acetylcholine esterase inhibitors (ng/L) in wastewater treatment plant (WWTP) effluent, and at upstream (US) and downstream (DS) locations of all 24 sites assessed in 2013 and 2014.** Box-Whisker plots with the line representing the median, the box the mean 50% of the data and the Whiskers the 10–90 percentile. Dots represent values outside this range. The black line indicates the annual average environmental quality standard (AA-EQS) value for the respective compound. The respective n is given in brackets after the compound name. Limits of quantification were determined individually for each sample and data are reported in detail in [34].

(section S8.4, Table O and S3_Data). Results of the two bioassays (YES, ERα-CALUX) were significantly correlated, however, ERα-CALUX generally measured higher $EEQ_{bio}$ values than YES (Fig C in S1 Appendix). The YES assay did not show exceedances of the assay specific EBT of 0.88 ng/L (n = 24 sites).

**Photosystem II and algae growth inhibition.** With regard to PSII and growth inhibition the impact of the WWTP was considerable too. $DEQ_{bio}$ concentrations for PSII inhibition were significantly higher (mean: 126 ng/L) than in the river upstream of the WWTP (mean: 33 ng/L). Highest values were measured in the WWTP effluent (mean: 187 ng/L) (Fig 5A and Table P in S1 Appendix). $DEQ_{bio}$ for growth inhibition showed a similar picture with highest values measured in the WWTP effluent (mean: 1370 ng/L), followed by the downstream sites (mean: 559 ng/L). Lowest values were measured upstream (mean: 284 ng/L) (Fig 5B and Table P in S1 Appendix).

For PSII inhibiting compounds exceedances of the corresponding EBT value (70 ng/L) [58, 59] were observed at 3 upstream and 7 downstream sites, whereas exceedances of the EBT value for growth inhibition (130 ng/L) [58] occurred at most of the 24 sites (18x upstream, 20x

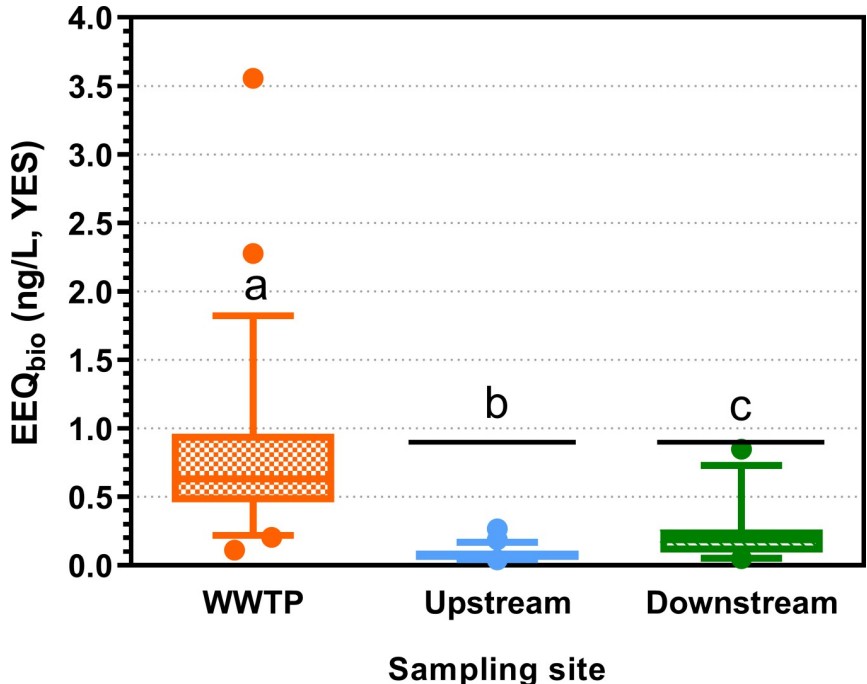

**Fig 4. Estrogenic activity in the Yeast Estrogen Screen: 17β-estradiol equivalent concentrations (EEQ$_{bio}$ in ng/L) at 24 sites investigated in 2013 and 2014 in wastewater treatment plant (WWTP) effluent as well as in the river up- and downstream of the WWTP discharge.** Box-Whisker plots with the line representing the median, the box the mean 50% of the data and the Whiskers the 10–90 percentile. Dots represent values outside this range (n = 24). Different letters indicate significant differences (Friedman test followed by Dunn's Multiple Comparison Test). Limits of quantification (LOQ) were determined for each sample and ranged from 0.03–0.18 ng/L EEQ$_{bio}$ for WWTP effluent and 0.01–0.08 ng/L EEQ$_{bio}$ for river samples. The black line represents the effect-based trigger value (0.88 ng/L EEQ$_{bio}$) [58]. Data are provided in S2_Data.

downstream) (see also Fig P in S1 Appendix). The strongest exceedance of both EBTs was observed at site #1 downstream (Buttisholz) with the highest DEQ$_{bio}$ concentration by far (1576 ng/L PSII DEQ$_{bio}$, 3845 ng/L growth DEQ$_{bio}$) (see also Fig Q in S1 Appendix).

DEQ$_{bio}$ values for PSII inhibition after 2 h and for growth inhibition after 24 h were highly correlated (Fig 6A). The slope of the regression being lower than 1 indicates a non-linear relationship between the two endpoints. When plotting 2 h PSII EC$_{50}$ values against 24 h growth EC$_{50}$ values (Fig 6B), more than half of the 2 h PSII inhibition data lie above the 1:1 line. In these samples the endpoint PSII inhibition after 2 h was more sensitive than the endpoint growth inhibition after 24 h indicating that PSII inhibiting herbicides dominate algae toxicity, as also observed in earlier studies [2, 62].

**Acetylcholinesterase inhibition.** The strongest inhibition of the enzyme acetylcholinesterase was detected in WWTP effluent samples (mean: 1249 ng/L PtEQ$_{bio}$). Lowest PtEQ$_{bio}$ concentrations were measured in the river upstream of the WWTP (mean$_{upstream}$: 249 ng/L), and river concentrations were elevated below the WWTP (mean$_{downstream}$: 411 ng/L) (Fig 7 and Table Q in S1 Appendix).

PtEQ$_{bio}$ values resulted in exceedances of the EBT (196 ngL) at the majority of up- and downstream sites (16x upstream, 18x downstream, n = 24 sites) (see black lines in Fig 7). However, these results need to be interpreted with care because of potential interference by other water constituents with the assay (see discussion).

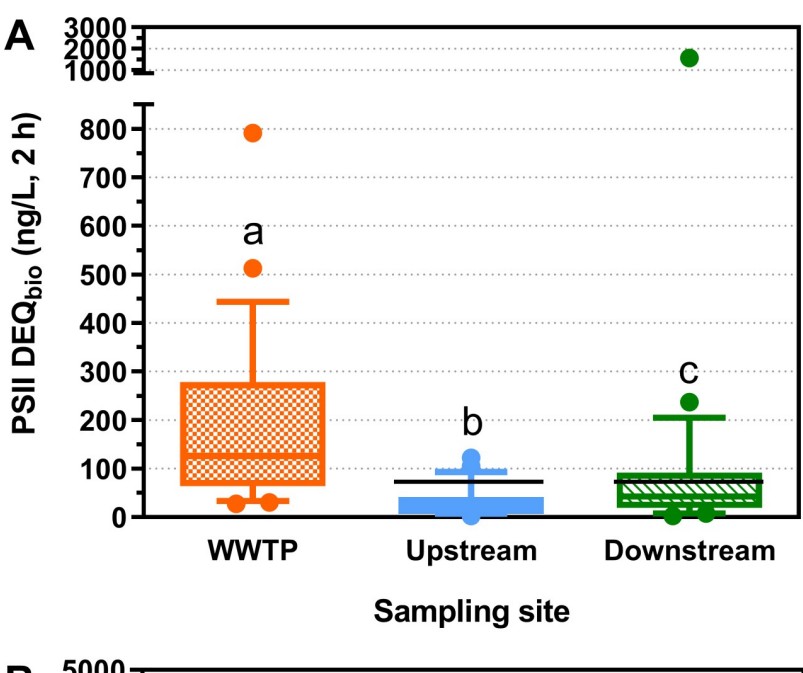

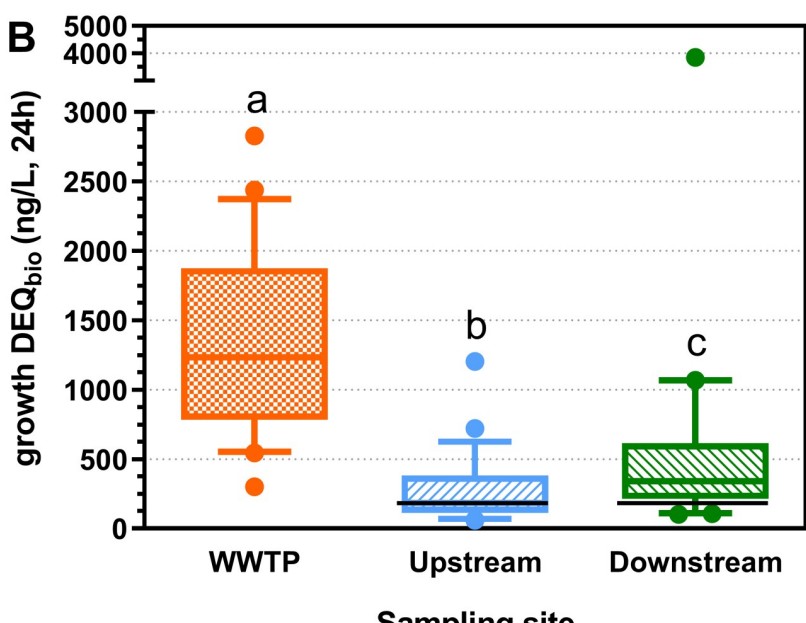

**Fig 5. Photosystem II and growth inhibition in Raphidocelis subcapitata: Diuron equivalent concentrations (DEQ_bio in ng/L) for (A) photosystem II and (B) growth inhibition at 24 sites investigated in 2013 and 2014 in wastewater treatment plant (WWTP) effluent as well as in the river upstream and downstream of the WWTP discharge (n = 24).** Box-Whisker plots with the line representing the median, the box the mean 50% of the data and the Whiskers the 10–90 percentile. Dots represent values outside this range. Different letters indicate significant differences (Friedman test followed by Dunn's Multiple Comparison Test). Limits of quantification (LOQs) were determined for each sample and ranged from 2–12 ng/L for WWTP effluent and 1–6 ng/L for river samples (2 h DEQ_bio, PSII inhibition). LOQs for 24 h growth inhibition ranged from 35–360 and from 70–720 ng/L DEQ_bio for river and WWTP effluent samples respectively. The black lines represent the effect-based trigger values for 2 h PSII DEQ_bio (70 ng/L) and 24 h growth DEQ_bio (130 ng/L) [58]. Data are provided in S4_Data.

**Effects measured in *in vivo* bioassays.** In general, reproduction of *Ceriodaphnia dubia* was enhanced by the tested samples, leading to values above 100%. At one of the four

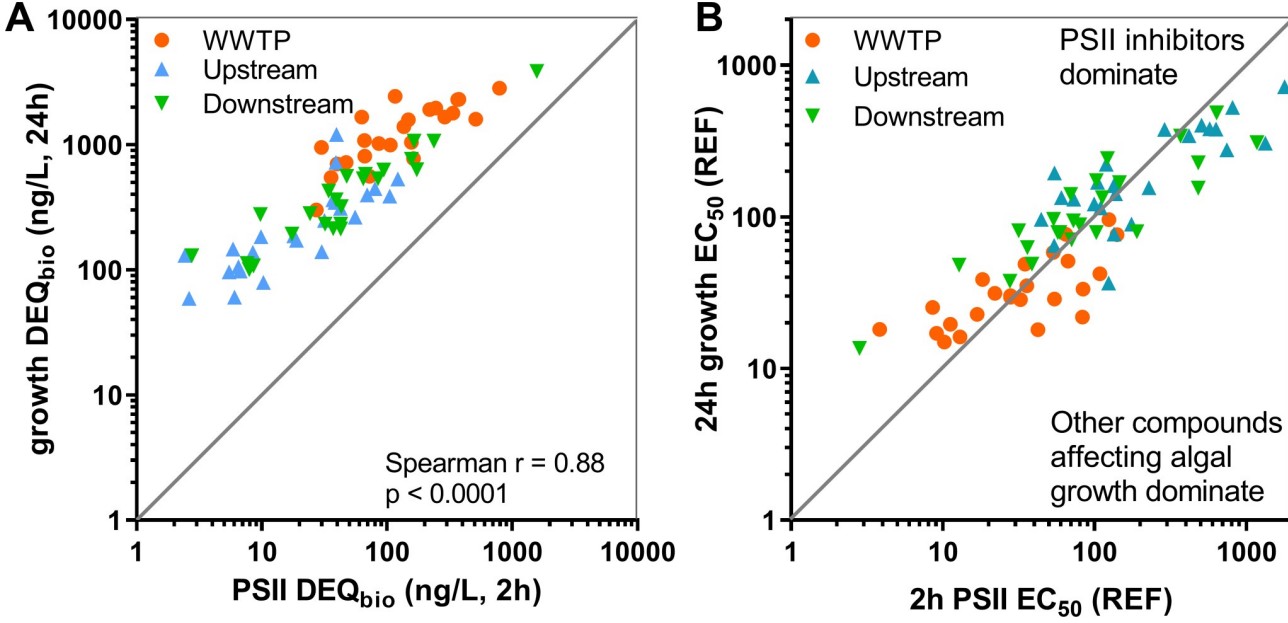

**Fig 6. Correlation of the different endpoints measured in the combined algae assay: (A)** diuron equivalent concentrations (DEQ$_{bio}$, ng/L) for PSII inhibition after 2 h and for growth inhibition after 24 h, **(B)** relationship between the EC$_{50}$ values of 24 h algae growth inhibition and 2 h photosystem II (PSII) inhibition measured in samples from 24 sites investigated in 2013/2014 in wastewater treatment plant (WWTP) effluent as well as in the river upstream and downstream of the WWTP discharge. Significant correlation for all values, as well as for the different sample groups (WWTP, upstream, downstream). Mean. n = 72. Data are provided in S4_Data and S5_Data.

investigated sites (site #6), reproduction downstream was significantly lower compared to upstream (Fig K in S1 Appendix). Feeding rate of *Gammarus fossarum* was significantly reduced downstream at one of four investigated sites (#1, Buttisholz) (Fig L in S1 Appendix). Further details on the *in vivo* bioassays can be found in section S8.7 in S1 Appendix.

**Correlation of bioassay results with chemical analysis.** For two bioassays (YES and combined algae assay), BEQ values measured in the bioassays (BEQ$_{bio}$) were significantly correlated to BEQs calculated with chemical-analytical data (BEQ$_{chem}$) taking into account the respective relative potencies of the measured compounds (see Tables F, G, and H in S1 Appendix), while in one bioassay BEQ$_{chem}$ underestimated BEQ$_{bio}$ (AChE inhibition assay) (Fig 8).

*Estrogenic activity*: EEQ$_{chem}$ was mostly below EEQ$_{bio}$ in the low concentration range (Fig 8A). In a few samples with high concentrations, EEQ$_{chem}$ was higher than EEQ$_{bio}$. On average, 80% of effects measured in the YES were explained by the estrogens analysed, with estrone contributing 56%, 17β-estradiol 23%, bisphenol A 0.4% and nonylphenol 0.3% (Fig 9). This picture changed partly, when incorporating chemical-analytical data below the LOQ as LOQ/2. In this additional analysis we assumed that compounds below LOQ contributed to the effect at half of their respective LOQ levels as suggested and discussed for estrogenic compounds by Kase, Javurkova [63]: The correlation of EEQ$_{bio}$ and EEQ$_{chem}$ improved (see Fig H in S1 Appendix). This might indicate compounds below LOQ playing a role in the bioassays, but, in most cases, led to an overestimation of the estrogenic activity (mean: 132%) and the contribution of 17α-ethinylestradiol to the overall activity became substantial (accounting for about 37%) due to its high estrogenic potency (see Fig I in S1 Appendix). However, it has to be taken into account that the method is uncertain in the range of the LOQ and, that a general LOQ was used across all samples for each measured compound. Contrarily, individual LOQs were determined for each sample for PSII inhibitors and insecticides.

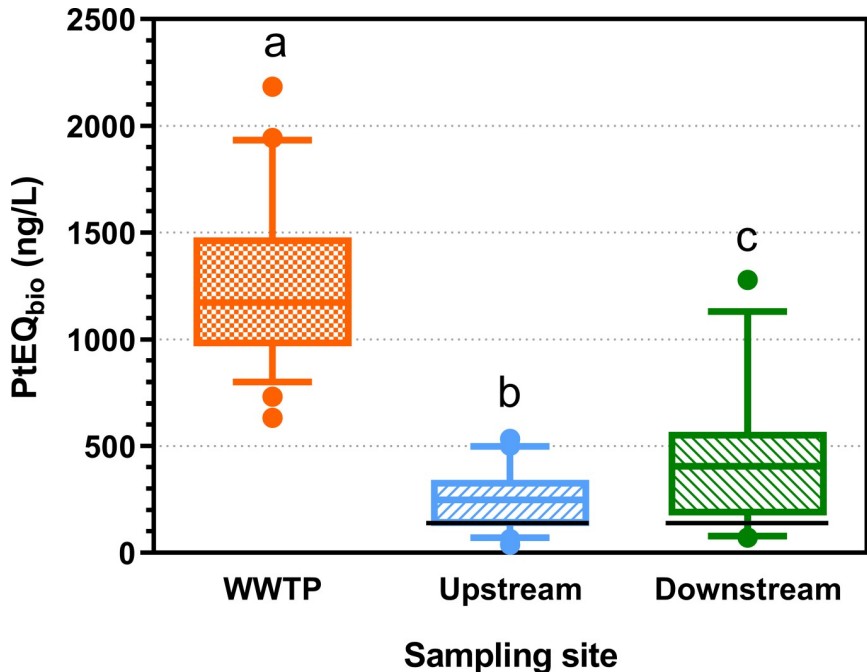

**Fig 7. Acetylcholinesterase inhibition: Parathion equivalent concentrations (PtEQ$_{bio}$, ng/L) at 24 sites investigated in 2013 and 2014 in wastewater treatment plant (WWTP) effluent as well as in the river up- and downstream of the WWTP discharge.** Box-Whisker plots with the line representing the median, the box the mean 50% of the data and the Whiskers the 10–90 percentile. Dots represent values outside this range (n = 24). Different letters indicate significant differences (Friedman test followed by Dunn's Multiple Comparison Test). Limits of quantification (LOQ) were determined for each sample and ranged from 43–532 ng/L PtEQ$_{bio}$ for WWTP effluent and 22–272 ng/L PtEQ$_{bio}$ for river samples. The black line represents the effect-based trigger value for PtEQ$_{bio}$ (196 ng/L). Data are provided in S6_Data.

*Algae PSII inhibition*: A very good correlation between the bioassay and chemical-analytical results occurred with most values lying around the 1:1 line (Fig 8B). Concentrations of quantified PSII inhibiting herbicides explained a high percentage of the effects seen in the combined algae assay, in average 81% in the WWTP effluent, 92% in the upstream river samples and 90% in the downstream river samples (Fig 10). In WWTP effluent, diuron, metribuzin, terbutryn and terbuthylazine, with mean contributions of 50, 8, 12 and 8%, respectively, explained the highest part of the measured effects. The picture changed when looking at up- and downstream sites in the river. Depending on the sites, the importance of diuron (mean US: 34%, DS: 38%) and terbutryn (mean US: 7%, DS: 11%) decreased, and the importance of lenacil (mean US: 8%, DS: 6%), linuron (mean DS: 4%), metribuzin (mean US: 7%, DS: 7%), terbutylazine-2-hydroxy (mean US: 15%, DS: 9%) and terbuthylazine (mean US: 20%, DS: 16%) increased. Only minor differences were observed with regard to the correlation of DEQ$_{bio}$ and DEQ$_{chem}$ with and without data below LOQ as LOQ/2 (see Figs H and J in S1 Appendix). A high correlation of DEQ$_{bio}$ and DEQ$_{chem}$ values has also been reported in earlier studies [2, 64].

The applied approach has a few limitations: For one of the most important herbicides, terbuthylazine, the main compound as well as its metabolite (terbuthylazine-2-hydroxy) were detected. The main compound explained up to 57% of the observed effects in WWTP effluents, up to 80% at individual upstream and up to 64% at individual downstream sites. In the river samples, also terbuthylazine-2-hydroxy gained importance and explained up to 36% (upstream) and 22% (downstream) of the observed effects at individual sites. As only the relative potency of the main compound was determined (see Table G in S1 Appendix), this value

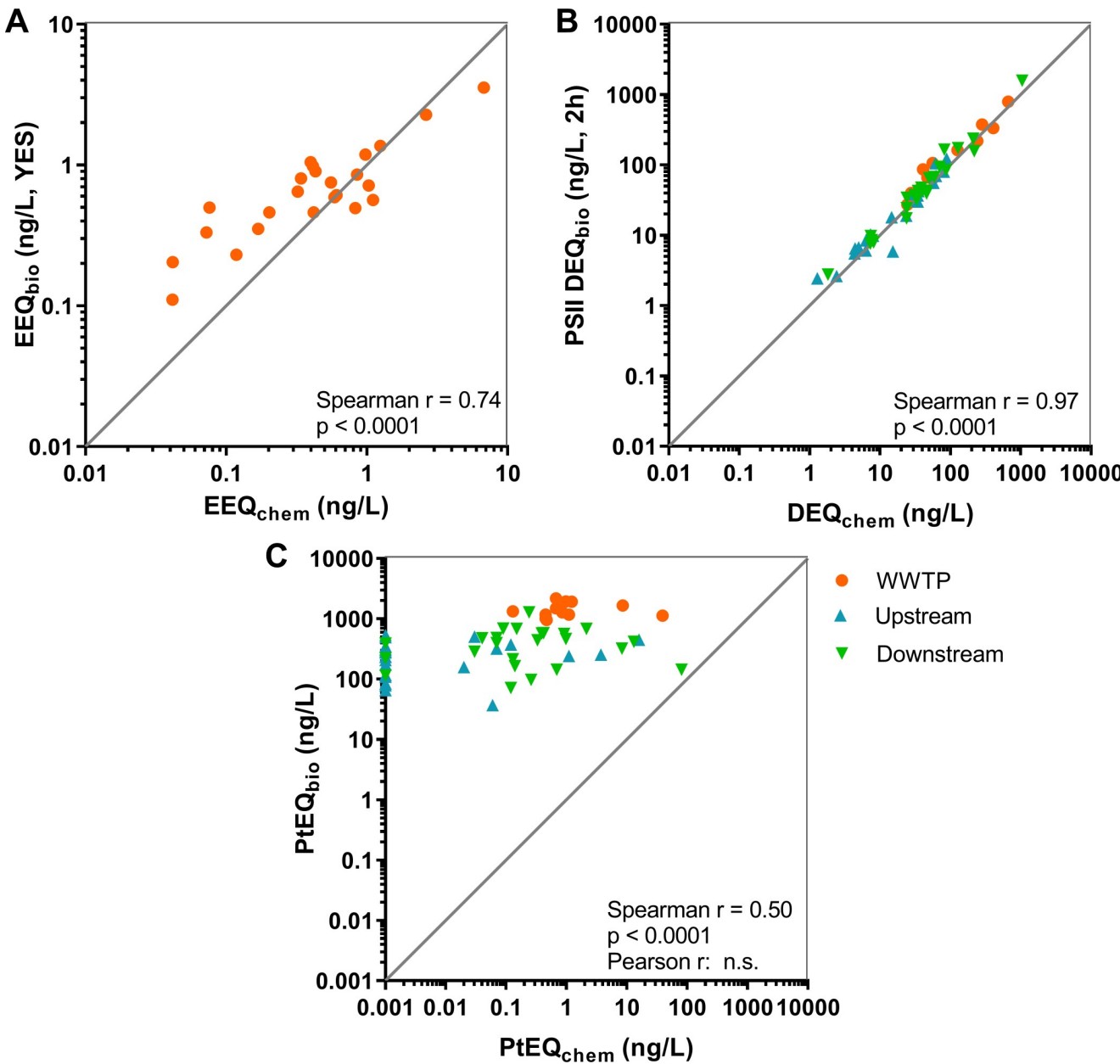

**Fig 8. Correlation of bioanalytical equivalent (BEQ$_{bio}$) concentrations measured in (A) the Yeast Estrogen Screen (YES, 17β-estradiol equivalent concentrations, EEQ$_{bio}$, ng/L), (B) the combined algae assay (diuron equivalent concentrations, DEQ$_{bio}$, ng/L), and (C) the acetylcholinesterase (AChE) inhibition assay (parathion equivalent concentrations, PtEQ$_{bio}$, ng/L) to the values calculated by chemical analysis (EEQ$_{chem}$, DEQ$_{chem}$, PtEQ$_{chem}$ resp.) based on relative potencies of the measured estrogens, PSII inhibitors or AChE inhibitors in the bioassays.** BEQs at 24 sites investigated in 2013/2014 in wastewater treatment plant effluent (all bioassays) as well as in the river up-and downstream of the WWTP discharge (combined algae assay and AChE inhibition assay). N = 24 (YES), n = 60 (algae, AChE). For the AChE assay "zero" values are displayed at 0.001 ng/L on the x-axis.

was also applied for calculating the relative contribution of the metabolite. It has to be taken into account that this is speculative and still needs to be confirmed in future studies. In addition, the different extraction methods present a confounding factor in these comparisons. It was recently found that metribuzin is well extracted with the SPE method used for chemical

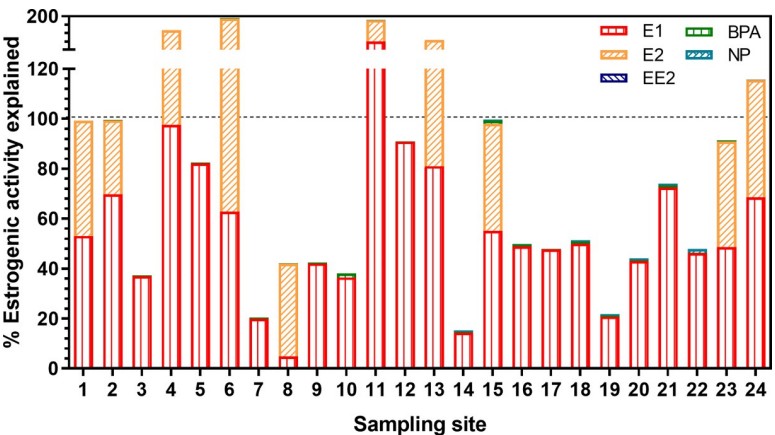

**Fig 9. Relative contribution of individual estrogens measured in chemical analysis (EEQ$_{chem}$, ng/L, calculated based on relative potencies of the measured estrogens in the bioassay) to the 17β-estradiol equivalent concentrations (EEQ$_{bio}$, ng/L) measured in the Yeast Estrogen Screen (YES).** Mean values from wastewater treatment plant effluent of 24 sites investigated in 2013 and 2014. E1 = estrone, E2 = 17β-estradiol, EE2 = 17α-ethinylestradiol, BPA = bisphenol A, NP = nonylphenol. Limits of quantification (LOQ) were 0.1 ng/L (E1), 0.2 ng/L (E2), 0.3 ng/L (EE2), 1.6 ng/L (BPA) and 1.2 ng/L (NP).

analysis, but gets largely lost in the SPE method used for bioassays (recovery range: ca. 10–60% [48]). This also required further work.

*AChE inhibition*: PtEQ$_{bio}$ and PtEQ$_{chem}$ correlated only if calculated with a non-parametric model (Spearman), but not when using a parametric one (Pearson) (Fig 8C). The bioassay indicates effects that are 4 to 5 orders of magnitude larger than the PtEQ$_{chem}$ values. This holds true especially for low PtEQ$_{chem}$ observations. Generally, the PtEQ$_{bio}$ values are within a narrow range (mostly within one order of magnitude for each location) while the PtEQ$_{chem}$ values cover a range of 2–4 orders of magnitude. The distribution of the PtEQ$_{chem}$−PtEQ$_{bio}$ data pairs suggest low sensitivity of the bioassay in the range of the quantified insecticide concentrations.

Only few of the 12 compounds for which relative potencies had been determined were detected above their respective LOQ, explaining little of the detected effect (WWTP effluent: maximum 3.5%, with chlorpyrifos-methyl and diazinon being relevant, but at a very low level; upstream: maximum 3.5%, with carbofuran, chlorpyrifos, and diazinon being relevant; downstream: maximum 57%, carbofuran most relevant) (Fig 11A, 11C and 11E). When incorporating non-detected carbamate and organophosphate insecticides as LOQ/2 (Fig 11B, 11D and 11F), 2–20% of the effects were explained in WWTP effluent, 7–180% at upstream and 3–93% at downstream sites. In this approach, aldicarb, azamethiphos, carbofuran, chlorpyrifos and chlorpyrifos-methyl contributed most to the observed AChE inhibition in WWTP effluent. In addition to these compounds, methomyl played a role at the up- and downstream sites. Considerable differences in the extent of explained effect between experimental years 2013 (sites 1–12) and 2014 (sites 13–24) can largely be attributed to differences in the compounds analysed, their respective LOQs as well as the sampling frequency (from sampled twice to sampled six times per experimental year) (see Table B in S1 Appendix for an overview). The fact that effects were low at the 2014 sites, where lower LOQ values were achieved, suggests that non-detected compounds did not play a major role. However, the substantial difference between PtEQ$_{bio}$ and PtEQ$_{chem}$ remained, and this was mostly due to dissolved organic carbon concentrations (see discussion).

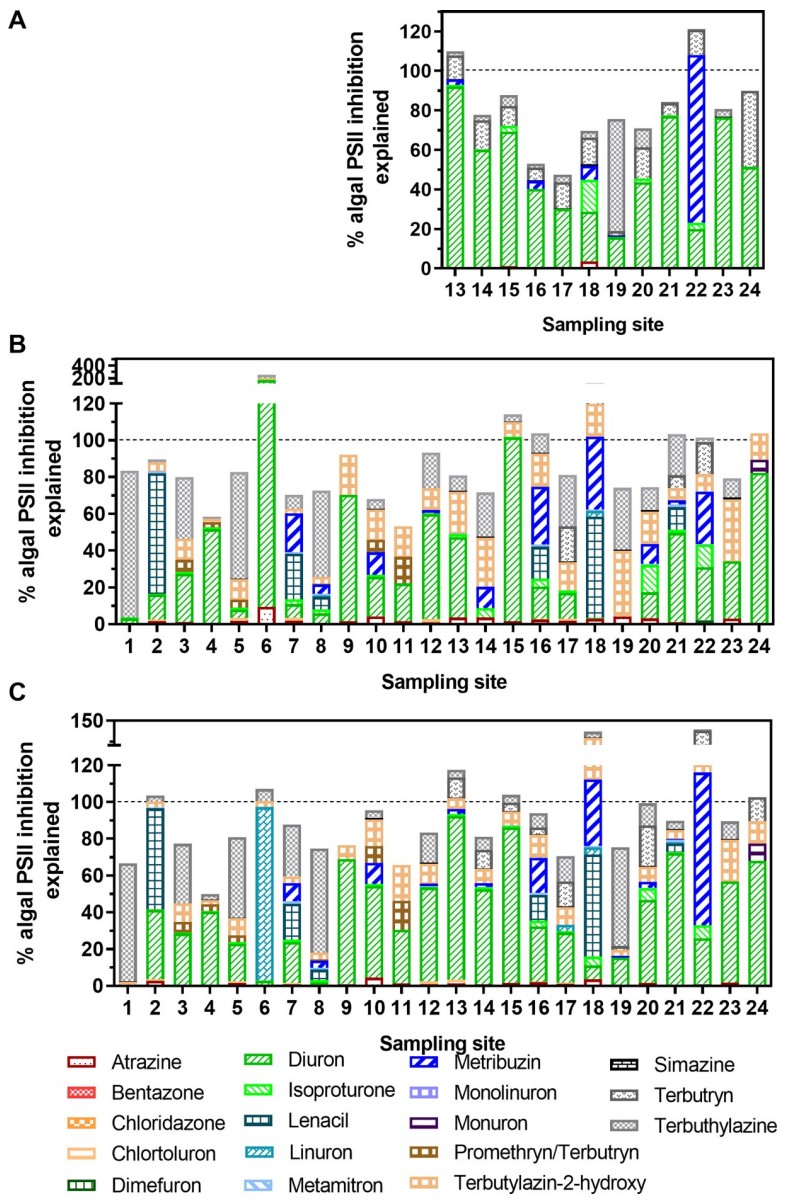

**Fig 10. Contribution of individual PSII inhibitors measured in chemical analysis (DEQ$_{chem}$, ng/L, based on relative potencies of the measured PSII inhibitors in the bioassay) to the diuron equivalent concentrations (DEQ$_{bio}$, ng/L) measured in the combined algae assay.** Mean values from wastewater treatment plant effluent of 12 sites (A: WWTP effluent) and 24 sites (B: upstream, C: downstream). Limits of quantification (LOQ) were determined individually for each sample and data are reported in detail in [34].

**Correlation of effects measured in effluents and river water.** *Estrogenic activity*: Effect measurements in WWTP effluent were predictive of estrogenicity at downstream sites, indicating that a majority of estrogenic compounds in the river water came from the WWTP: EEQ$_{bio}$ values measured at the downstream sites correlated significantly with EEQ$_{bio}$ values extrapolated from values measured in WWTP effluent using the respective dilution factor in the river (see Table C in S1 Appendix) (Spearman r = 0.85, p<0.0001) (Fig M-A in S1 Appendix). This was also the case for EEQ$_{bio}$ values from the ERα-CALUX (Spearman r = 0.72, p = 0.0128) (Fig M-B in S1 Appendix). In addition, the fraction of wastewater calculated based

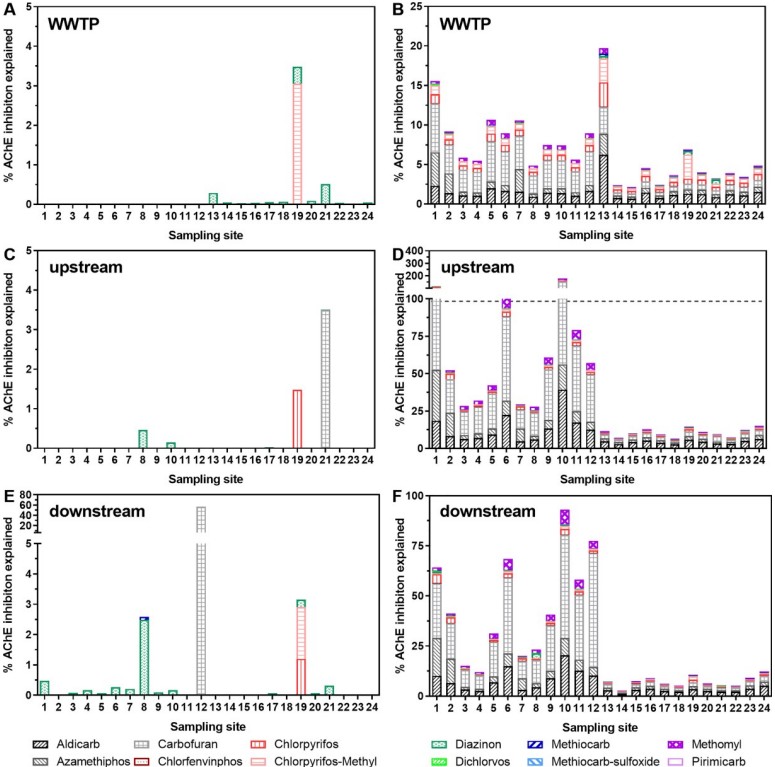

**Fig 11. Contribution of individual insecticides (%) measured in chemical analysis (PtEQ$_{chem}$, ng/L, calculated based on relative potencies of the measured insecticides in the bioassay) to the parathion equivalent concentrations (PtEQ$_{bio}$) measured in the acetylcholinesterase inhibition assay.** Mean values from wastewater treatment plant effluent (A, 12 sites, B 24 sites) and from samples taken upstream (C and D) and downstream (E and F) of the WW discharge at 24 sites investigated in 2013 and 2014. Limits of quantification (LOQ) were determined individually for each sample and are reported in detail in [34]. A, C, and E: only compound concentrations above LOQ included, B, D and F: compound concentrations above LOQ plus LOQ/2 concentrations (where no concentrations above LOQ were measured). For WWTP effluent, LOQs were not available for all relevant compounds. In this case the LOQ of the next similar sample type was used, i.e. the one from the downstream sampling site. Differences between samples 1–12 (2013) and 13–24 (2014) are largely related to differences in analysed compounds, their respective LOQs as well as sampling frequency.

on water quality parameters such as conductivity, pH, alkalinity, hardness, nutrients and major ions was significantly correlated with the fraction of wastewater calculated based on bioassay results (YES: Spearman r = 0.79, p < 0.0001; ERα-CALUX: Spearman r = 0.81, p = 0.0025) (Fig M-C and -D in S1 Appendix).

Algae *photosystem II and growth inhibition*: With regard to PSII inhibiting compounds, DEQ$_{bio}$ values for PSII inhibition measured at downstream sites were significantly correlated to values calculated based on DEQ$_{bio}$ measured in WWTP effluent and the respective dilution factors in the river (Spearman r = 0.99, p<0.0001) (Fig N-A in S1 Appendix). Similar results were obtained for growth inhibition (Spearman r = 0.90, p<0.0001) (Fig N-B in S1 Appendix). The fraction of wastewater calculated based on water quality parameters was significantly correlated to the fraction of wastewater calculated based on bioassay results for the endpoints PSII inhibition (Spearman r = 0.96, p < 0.0001) and growth inhibition (Spearman r = 0.51, p = 0.02) (Fig N-C and -D in S1 Appendix). As the highest downstream value (site #1) was not related to input from WWTP effluent but to a herbicide discharge between the WWTP effluent and the downstream sampling site in the river, outliers were excluded for this analysis.

*AChE inhibition*: $PtEQ_{bio}$ values measured in the river were highly correlated to values extrapolated based on $PtEQ_{bio}$ in WWTP effluent and the dilution factors (Spearman r = 0.98, p<0.0001). Similarly, the fraction of wastewater calculated based on water quality parameters was significantly correlated to the fraction of wastewater calculated based on bioassay results (Spearman r = 0.89, p<0.0001) (Fig O in S1 Appendix).

## Discussion

Earlier studies showed that elevated concentrations of micropollutants in wastewater-impacted streams can elicit serious ecotoxicological effects on aquatic organisms such as a reduced capacity to photosynthesise and grow (plants), reproduction and neurotoxic effects (invertebrates and vertebrates) which may lead to changes in behaviour and / or community composition (e.g. [2–4, 7–9, 26]). Several studies have demonstrated ecotoxicological effects *in vitro*, *in vivo* and *in situ* in single streams and were able to partly identify the responsible compounds (e.g. [25, 29–31, 65]. However to date, a comprehensive, large-scale field study on the effects of micropollutants has been lacking. The current study, which was performed at 24 Swiss WWTP and in adjacent streams as part of the project EcoImpact [33], aimed at closing this gap. We focused on the *in vitro* ecotoxicological assessment of estrogenicity (YES and ERα-CALUX), algal toxicity (combined algae assay), and neurotoxicity (AChE inhibition assay) complemented with *in vivo* assessment of water flea reproduction and *in situ* amphipod feeding assays at selected sites. We evaluated how much of the observed effects determined in *in vitro* bioassays were explained by results of chemical analyses, and we performed a risk assessment based on chemical analysis as well as bioassay results. In general, all four endpoints measured by *in vitro* bioassays proved to be robust. Selected *in vivo* assays provided valuable information on decreased water flea reproduction and amphipod feeding at one of four assessed sites respectively.

Our results showed that concentrations of estrogenic compounds, PSII inhibiting herbicides and carbamate and organophosphate insecticides were either lower or in the range of previous studies. For estrogenic activity, river values measured downstream of the WWTP in the current study ranged from 0.05 to 0.85 ng/L $EEQ_{bio}$, while Vermeirssen, Burki [46] reported concentrations in 18 Swiss rivers ranging from 0.3 to 2 ng/L (during winter) and from 0.4 to 7 ng/L (during summer) [46]. However, the older study focused on hotspots which could explain the higher activities measured. Rivers downstream of 14 Swiss WWTPs analysed by Kienle, Kunz [66] contained 0.1 to 5.5 ng/L $EEQ_{bio}$ which was significantly higher than at upstream sites (<LOQ to 1.8 ng/L $EEQ_{bio}$); effluents in this study contained 0.6 to 11 ng/L $EEQ_{bio}$, whereas effluents in our study contained 0.11 to 3.6 ng/L. In a recent EU-wide study, again targeting hotspots, $EEQ_{bio}$ were 0.03 to 23 ng/L (n = 17) in WWTP effluent, and 0.06 to 1.2 ng/L (n = 16) in river water samples. In this study, the ERα-CALUX assay was used to measure estrogenic activity [67]. The values for estrogenic compounds determined by chemical analysis in the current study are also lower than in other studies on Swiss as well as other European WWTP effluents [18, 67, 68].

Our results show that WWTP are an important point source of herbicidal compounds. This is confirmed by previous studies using *in vitro* bioassays to detect PSII inhibiting compounds, which focused predominantly on wastewater. $DEQ_{bio}$ values for PSII inhibition in secondary-treated effluent of Australian WWTPs were 83, 200 and 242 ng/L [2]. At WWTP Wüeri (Regensdorf, Switzerland), in average, 160 ng/L $DEQ_{bio}$ were measured [49] and 228 ng/L $DEQ_{bio}$ at WWTP Vidy (Lausanne, Switzerland) [18]. Data for surface waters are, however, relatively scarce. Escher, Bramaz [49] report average $DEQ_{bio}$ of 230 ng/L and 190 ng/L in water samples collected from the Furtbach at WWTP Wüeri (with and without effluent

respectively). $DEQ_{bio}$ values in the present study ($mean_{WWTP}$ = 187 ng/L; $mean_{upstream}$ = 33 ng/L, $mean_{downstream}$ = 127 ng/L) were in a similar range. Interestingly, at the site where the highest $DEQ_{bio}$ (and also $DEQ_{chem}$) values were measured (1576 ng/L $DEQ_{bio}$, Buttisholz), feeding activity of amphipods was significantly decreased compared to the upstream site (see Fig K in S1 Appendix); these high values were caused by a discharge of the herbicide terbuthylazine between the WWTP effluent and the downstream sampling site in the river.

Levels of AChE inhibition measured in this study were relatively high with mean $PtEQ_{bio}$ values of 1249 ng/L (WWTP effluent), 249 ng/L (upstream) and 411 ng/L (downstream), but in the range of earlier studies. Escher, Bramaz [49] reported average $PtEQ_{bio}$ values of 510 ng/L in WWTP effluent (secondary treatment) in Switzerland and 270 and 210 ng/L in river water (with and without WWTP effluent, respectively). However, the measured values were not confirmed by chemical-analytical results, and should be interpreted with caution as dissolved organic carbon can influence assay results.

### Dissolved organic carbon affects results of the AChE inhibition assay

It is known that abiotic parameters, such as dissolved organic carbon or pH, can influence the results of *in vitro* bioassays. While this effect is negligible with regard to the combined algae assay [69], it is a confirmed confounding factor for the AChE inhibition assay used in this study [70]. No prior information was available for the YES assay. Neale and Escher [70] report a suppressive effect of co-extracted dissolved organic carbon on AChE activity (i.e. higher $PtEQ_{bio}$ values) at dissolved organic carbon concentrations $> 2$ mg $L^{-1}$. (measured in SPE extracts). In our study, dissolved organic carbon concentrations in native samples were mostly higher than 2 mg $L^{-1}$ (see SI section S8.1 and S7_Data). Assuming an extraction efficiency for dissolved organic carbon of 40% in WWTP effluent and of 70% in river water, as measured by [70], in the current study, 13 WWTP effluents, 5 upstream and 14 downstream river samples (of 24 samples each) would exceed this concentration. An impact of dissolved organic carbon on results of the ACHE inhibition assay was confirmed in the current study: While a correlation of $BEQ_{bio}$ values with dissolved organic carbon concentrations occurred for all three *in vitro* assays (Spearman r = 0.72, 0.58 and 0.69 for $PtEQ_{bio}$, $DEQ_{bio}$, and $EEQ_{bio}$ values (YES), $p < 0.0001$) (see Fig G in S1 Appendix), a multiple regression revealed that, with regard to the variability of results, DOC only played a role for $PtEQ_{bio}$ values: 45% of the variability was jointly explained by dissolved organic carbon and $PtEQ_{chem}$. Individually, these factors accounted for 42% and 20% of the variance, respectively. On the other hand, variability of $DEQ_{bio}$ values was strongly linked to $DEQ_{chem}$, which explained 96% of the variance as a single factor while dissolved organic carbon had hardly any explanatory value (4%). For $EEQ_{bio}$ values, for which no information on potential interferences with dissolved organic carbon was available from the literature, 76% of the variability was explained by $EEQ_{chem}$. A joint regression including dissolved organic carbon did not improve the explanatory power (75% for the multiple regression) in this case. This indicates that dissolved organic carbon concentrations do not have a strong influence on $EEQ_{bio}$ values. However, to refine these findings, further experiments complementing those performed by Neale and Escher [70] and Neale and Escher [69] are necessary. To overcome effects of confounding factors on AChE inhibition, it can also be assessed as biomarker directly in organisms (for review see [71] and [72]).

### Risk assessment based on single chemicals partly underestimates the mixture risk as well as the risk identified by ecotoxicological bioassays

*Risk of estrogenic compounds*: Chemical-analytical data for individual estrogenic compounds in WWTP effluents and extrapolated to river water using respective dilution factors, revealed

few exceedances of water quality thresholds (AA-EQS) at downstream sites (estrone, 17β-estradiol: 1x (n = 24 sites); nonylphenol 2x (n = 12 sites)). Assessment of the mixture risk for all estrogens measured ($RQ_{mix}$), by summing up the individual risk quotients (ratio between measured environmental concentration and AA-EQS), revealed exceedances of the $RQ_{mix}$ at 6 of 24 sites. This clearly demonstrates that the ecological risk of estrogenic substances is under-estimated when only single compounds are considered, as previously highlighted for fungi-cides, insecticides and pesticides [11].

The risk assessment based on effect data obtained by the YES or ERα-CALUX goes a step further because effect-based (or bioanalytical) tools additionally measure activities of com-pounds not quantified, not quantifiable or unknown [63, 67]. Here the choice of the assay-spe-cific effect-based trigger value (EBT) plays an important role. When using an EBT of 0.88 ng/L $EEQ_{bio}$ for the YES, as recommended by Escher, At-Assa [58], no exceedances were found at any site (Fig 4) indicating no risk with regard to estrogenic compounds to aquatic organisms; applying an EBT of 0.1 ng/L $EEQ_{bio}$ for the ERα-CALUX [58] resulted in 5 exceedances at upstream and 11 exceedances at downstream sites (of 12 sites total) (Figs D and P in S1 Appen-dix). This higher number of exceedances is partly related to the substantially lower EBT for the ERα-CALUX, but it also has to be taken into account that this assay presented generally higher $EEQ_{bio}$ values than the YES. Its 10x higher sensitivity with regard to 17β-estradiol compared to the YES may also play a role, however of minor importance, as sensitivity differences for estrone, the dominant steroid in the sample, between both assays are smaller [56]. When using the AA-EQS of the reference compound 17β-estradiol (0.4 ng/L) as EBT, as suggested as a pragmatic approach by [59, 63, 73], this picture would again change to one exceedance at a total of 24 downstream sites for the YES and three exceedances at 12 downstream sites for the ERα-CALUX. The differences in the EBT values suggested by [58] for both assays are based on the fact that they include information on the relative potency of a number of estrogenic com-pounds in the assay as well as their respective AA-EQS. This approach is scientifically sound, but also renders the interpretation of the results more difficult, especially if, as in the current study, two bioassays with the same endpoint are resulting in a differing number of exceed-ances. Taking into account the relevance of *in vitro* responses in the ERα-CALUX for *in vivo* effects in zebrafish embryos, Brion, De Gussem [74] recently suggested an EBT value of 0.28 ng/L.

*Risk of PSII inhibitors*: Based on chemical-analytical data of stream water samples, AA-EQS exceedances occurred for 5 of 18 measured PSII inhibitors at single upstream and at 1–5 downstream sites; only the herbicides diuron and terbuthylazine exceeded its AA-EQS at more than 1 downstream site (i.e. 5 and 2 respectively). $RQ_{mix}$ for PSII inhibitors was >1 at 4 upstream and 15 downstream sites. This indicates that significant amounts of PSII inhibiting compounds are released via WWTP effluent.

The effect-based risk assessment based on results of the combined algae assay resulted in fewer exceedances of the EBT for PSII inhibition (70 ng/L $DEQ_{bio}$ [58, 59]) (3x upstream and 7x downstream) than the $RQ_{mix}$ approach based on chemical analysis (4x upstream and 15x downstream). A potential reason for this might be the partial loss of certain PSII inhibiting compounds during the extraction procedure, such as metribuzin and metamitron. As men-tioned above, the SPE applied to produce extracts tested in bioassays is different from the SPE used for extraction of chemical analyses [48]. The EBT for growth inhibition of 130 ng/L $DEQ_{bio}$ [58] was exceeded at the majority of sites (18 upstream, 20 downstream). This might be due to other compounds affecting algae growth, such as metazachlor [75]. However, $DEQ_{bio}$ for growth inhibition are somewhat less robust than $DEQ_{bio}$ for PSII inhibition, mainly due to limitations in the experimental setup (e.g. a suboptimal growth in the wells compared to bigger vessels) [47] and thus a higher variability in the obtained results.

*Risk of AChE inhibitors*: Risk assessment based on chemical analysis revealed that individual AA-EQS of six of the 14 measured AChE inhibitors were exceeded. This occurred, as for PSII inhibitors, mostly at single sites. An exception was diazinon, which, with AA-EQS exceedances at 11 downstream sites, was the compound presenting the highest risk for aquatic organisms as also discussed by [34]. $RQ_{mix}$ for AChE inhibitors was exceeded at 2 upstream and 13 downstream sites. Downstream also stronger changes in the macroinvertebrate community composition were observed compared to upstream: The Species at Risk (SPEAR)-index indicated a significant loss of pesticide-sensitive species at downstream locations compared to upstream sites with a trend for a stronger decline with a higher fraction of WW downstream [35]. However, upstream land use, especially intensive agriculture, happened to be much more important for this index than the contribution of wastewater. The extent of intensive agriculture was a dominant driver for upstream macroinvertebrate communities, and, hence, indirectly for the communities downstream of the WWTP.

## The percentage of observed effect explained by chemical analysis is highest in the combined algae assay

For estrogenic compounds, $EEQ_{chem}$ explained on average 80% of $EEQ_{bio}$ determined by the YES (Fig 9), but only 12% of $EEQ_{bio}$ determined by the ERα-CALUX (Fig F in S1 Appendix). This can be largely explained by the weaknesses in the chemical analysis of estrogenic compounds, especially 17α-ethinylestradiol for which LOQs are often too high. In addition, the relative potencies (RPs) of the main estrogenic compounds to the reference compound 17β-estradiol are different between the YES and ERα-CALUX assays (Table F in S1 Appendix). Estrone concentrations ($RP_{YES}$ = 0.26, $RP_{ERα-CALUX}$ = 0.02) explained 56%, on average, of $EEQ_{bio}$ measured by YES, but only 2% measured by ERα-CALUX, while 17β-estradiol explained 23% (YES) and 10% (ERα-CALUX) of the detected estrogenic activity, on average. bisphenol A and nonylphenol played a minor role. The contribution of the most potent estrogen, 17α-ethinylestradiol ($RP_{YES}$ = 1.2, $RP_{ERα-CALUX}$ = 1.3), was not evaluated, because concentrations were always below the LOQ of the chemical analytical method.

Measured PSII inhibitors explained a large part of the $DEQ_{bio}$ values for PSII inhibition (Fig 10), as also observed in previous studies [2, 64]. In several cases, where $DEQ_{chem}$ was higher than $DEQ_{bio}$, metribuzin was a key player. This compound is only partially extracted by the SPE used for bioassays [48] while the SPE used for chemical analysis extracts this compound more efficiently. The correlation between $DEQ_{bio}$ and $DEQ_{chem}$ could therefore be improved further if the same extraction methods were used. In another sample (site 6, upstream), $DEQ_{chem}$ was higher than $DEQ_{bio}$: measured diuron concentrations accounted for 165% of the $DEQ_{bio}$ determined by bioassay. In this case, the relatively low $DEQ_{bio}$ value (5.9 ng/L) was in the range of the bioassay's LOQ (1.7 ng/L).

Measured concentrations of AChE inhibitors explained less than 4% of the $PtEQ_{bio}$ values (Fig 11), on average. There was only one sample which contained carbofuran at concentrations that explained 57% of the $PtEQ_{bio}$ measured by bioassay. The discrepancy could be due to partly high LOQ for AChE inhibitors in the chemical analysis, unknown AChE inhibiting compounds present in the samples, or the influence of dissolved organic carbon on bioassay results (see above).

Adding compounds below the LOQ as LOQ/2 values to determine the $BEQ_{chem}$ increases the percentage of effect explained by analytical data, especially for AChE inhibitors, but also for estrogenic compounds. This approach was suggested and discussed by Kase, Javurkova [63] for estrogenic compounds. It has to be kept in mind, however, that the respective chemicals may not be present in the sample. Adding 17α-ethinylestradiol (which was always below

the LOQ) as LOQ/2, gives a lot of weight to this compound as both its relative potency (Fig I in S1 Appendix) and the LOQ of 0.3 ng/L are high. However, its co-occurrence with estrone and 17β-estradiol in surface waters is likely. Where $EEQ_{chem}$ (with 17α-ethinylestradiol added at LOQ/2) far exceeds 100% of $EEQ_{bio}$, it would have to be assumed that LOQ/2 is an overestimation of the real 17α-ethinylestradiol concentration present in a water sample. Applying this approach to $PtEQ_{bio}$ values for AChE inhibition clearly shows that its usefulness is highly dependent on the respective LOQ values. The list of measured compounds as well as the LOQ values for individual compounds differed partly between sites 1–12 and 13–24. This led to a much higher percentage of effect explained by chemical analytical data at sites 1–12, sometimes above 100%. For example, the percentage $PtEQ_{bio}$ explained by carbofuran concentrations (the most important insecticide in terms of detects in this study), was much higher at sites 1–12 studied in 2013 compared to sites 13–24 studied in 2014 (Fig 11). This is solely related to the higher LOQ at sites 1–12 (7 resp. 7.4 ng/L) compared to sites 13–24 (1.5 ng/L) (for further details on LOQs see [34]). This shows that the concept is fallacious if applied to a data set with large differences in LOQ values for the same compound and should not be used in such a case. For PSII inhibitors adding non-detected but relevant compounds as LOQ/2 changes results little, indicating that this assay accurately measures the combined effects of PSII inhibitors present in water samples.

## Conclusions

In 24 Swiss streams receiving WWTP effluent the contamination with estrogenic substances can be considered low. Low estrogenicity and no exceedances of the effect-based trigger (EBT) value for estrogenicity measured in the YES were detected during a two year study (2013–14). Results from YES generally correlated well with ERα-CALUX results and seemed to correlate better with chemical analytics (LC-MS/MS) than ERα-CALUX. However, this was highly influenced by the relative potency differences for estrone in the two bioassays (towards 17β-estradiol) and the analytical detection challenges of potent steroidal estrogens (i.e. 17β-estradiol and 17α-ethinylestradiol). The EBT for photosystem II inhibiting herbicides was exceeded at 3 upstream and 7 downstream sites, demonstrating that WWTP effluent was a major but not the sole source of these chemicals. The EBT for AChE inhibition, the main toxicity mechanism of carbamate and organophosphate insecticides, was exceeded at a majority of the sites (16x upstream, 18x downstream), this effect was largely due to dissolved organic carbon and, in part, to carbofuran concentrations.

Effects measured in the combined algae assay correlated best with chemical-analytical data, while results of chemical analysis often underestimated estrogenicity due to LOQs for 17α-ethinylestradiol being too high. Mixture assessment based on chemical data strongly underestimated acetylcholinesterase inhibition, largely due to an assay-specific artefact caused by dissolved organic carbon.

We conclude that two *in vitro* bioassays used in this study, the YES measuring estrogenicity and the combined algae test measuring PSII and growth inhibition, are well suited for the ecotoxicological assessment of river water quality. The AChE inhibition assay with purified enzyme, however, revealed substantial limitations. Future work should quantify the potential influence of dissolved organic carbon in this bioassay.

Overall, based on the applied bioassays as well as the measured compounds, photosystem II inhibiting herbicides posed the highest risk for aquatic algae and plants at up- and downstream reaches of the investigated streams. Estrogens played a minor role, and it was difficult to draw conclusions on effects of acetylcholinesterase inhibiting insecticides because of confounding factors in the applied bioassay. However, several exceedances of mixture risk quotients for

organophosphate and carbamate insecticides indicated a potential risk for aquatic organisms, especially invertebrates and vertebrates, at downstream sites. In addition, feeding activity of amphipods and reproduction of water flea might be impaired at downstream reaches due to micropollutants originating from the WWTP.

## Supporting information

**S1 Appendix.**
(PDF)

## Acknowledgments

We are grateful to Thomas Bucher for determining relative potencies in the acetylcholinesterase inhibition assay, as well as to Barbara Ganser, Sibylle Birrer, Sereina Gut, Beatrice Läuppi, and Janine Mazenauer (all Ecotox Centre) for performing a part of the bioassays and helping with sampling, to Nadzeya Homazva (Ecotox Centre) for performing estrogen analysis and to Nicole Munz and Barbara Spycher (Eawag) for providing data from chemical analysis for PSII inhibitors and organophosphate and carbamate insecticides.

## Author Contributions

**Conceptualization:** Cornelia Kienle, Etiënne L. M. Vermeirssen, Andrea Schifferli, Heinz Singer, Christian Stamm, Inge Werner.

**Data curation:** Cornelia Kienle, Andrea Schifferli.

**Formal analysis:** Cornelia Kienle, Andrea Schifferli, Heinz Singer, Christian Stamm.

**Funding acquisition:** Christian Stamm, Inge Werner.

**Investigation:** Cornelia Kienle, Etiënne L. M. Vermeirssen, Andrea Schifferli, Heinz Singer.

**Methodology:** Cornelia Kienle, Etiënne L. M. Vermeirssen, Andrea Schifferli, Heinz Singer, Christian Stamm, Inge Werner.

**Project administration:** Etiënne L. M. Vermeirssen, Christian Stamm, Inge Werner.

**Resources:** Cornelia Kienle, Andrea Schifferli, Christian Stamm, Inge Werner.

**Supervision:** Etiënne L. M. Vermeirssen, Christian Stamm, Inge Werner.

**Validation:** Cornelia Kienle, Etiënne L. M. Vermeirssen, Andrea Schifferli, Heinz Singer, Christian Stamm.

**Visualization:** Cornelia Kienle, Christian Stamm.

**Writing – original draft:** Cornelia Kienle.

**Writing – review & editing:** Cornelia Kienle, Etiënne L. M. Vermeirssen, Andrea Schifferli, Heinz Singer, Christian Stamm, Inge Werner.

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
