## [Decision Letter · Decision Letter 0]

11 Oct 2019

PONE-D-19-25149

Effects of treated wastewater on the ecotoxicity of small streams – unravelling the contribution of chemicals causing effects

PLOS ONE

Dear Dr. Kienle,

Thank you for submitting your manuscript to PLOS ONE. After careful consideration, we feel that it has merit but does not fully meet PLOS ONE’s publication criteria as it currently stands. Therefore, we invite you to submit a revised version of the manuscript that addresses the points raised during the review process.

The reviewers provide several suggestions for improving the manuscript that should be considered.

We would appreciate receiving your revised manuscript by Nov 25 2019 11:59PM. To enhance the reproducibility of your results, we recommend that if applicable you deposit your laboratory protocols in protocols.io, where a protocol can be assigned its own identifier (DOI) such that it can be cited independently in the future. For instructions see: http://journals.plos.org/plosone/s/submission-guidelines#loc-laboratory-protocols

We look forward to receiving your revised manuscript.

Kind regards,

James P. Meador

Academic Editor

PLOS ONE

1. In your Methods section, please provide additional information regarding the permits you obtained to sample water at the EcoImpact sites for this work. Please ensure you have included the following information: 1) Permits and approvals obtained for the work, including the full name of the authority that approved the study. If none were required, please explain why; 2) Whether the land accessed is privately owned or protected.

2. Thank you for including your competiong interests statement; "The authors have declared that no competing interests exist."

We note that one or more of the authors are employed by a commercial company:Swiss Centre for Applied Ecotoxicology.

Reviewers' comments:

Reviewer's Responses to Questions

**Comments to the Author**

1. Is the manuscript technically sound, and do the data support the conclusions?

Reviewer #1: Yes

Reviewer #2: Yes

2. Has the statistical analysis been performed appropriately and rigorously? 

Reviewer #1: Yes

Reviewer #2: Yes

3. Have the authors made all data underlying the findings in their manuscript fully available?

Reviewer #1: Yes

Reviewer #2: Yes

4. Is the manuscript presented in an intelligible fashion and written in standard English?

Reviewer #1: Yes

Reviewer #2: Yes

5. Review Comments to the Author

Reviewer #1: The manuscript by Kienle et al., entitled "Effects of treated wastewater on the ecotoxicity of small streams – unravelling the contribution of chemicals causing effects" describes a far-tanging and in depth study of the effects of treated wastewater effluent on small and mid-size streams in Switzerland. The study is comprehensive, presented with clarity and great detail and provides important comprehensive evidence for the effects of micropollutants released into the aquatic environment through the discharge of treated municipal wastewater effluent. The combination of detailed water chemistry, functional assays and analysis in vitro and in vivo across many streams is noteworthy and adds great strength to the conclusions. The manuscript is extremely well written and only requires minor edits to ready it for publications. Methods are exhaustingly described and referenced and Results are well characterized in the Discussion. Methods and especially the Results are extensively documented in the Supplemental Materials accompanying the main text. The manuscript is rather long and would benefit from "light trimming". The second to last paragraph of the Introduction (Lines 98 through 116) is more appropriate for the Discussion (where much of it is repeated already) and its purpose tom place the current study into the context of previous and complementary work could readily be accomplished in adding one sentence to the last paragraph of the discussion.

A minor issue, but one that effects the readability of the manuscript is the excessive use of abbreviations for terminology that is not used much throughout the manuscript. The authors appeared to have struggled with the abundance of abbreviations as well and frequently switch from abbreviation (for example "MP", AChE, same with TP - which in the US stands for "toilet paper' - not what the authors meant, but somewhat appropriate in the context of the research :-) to the full word throughout the text. Spelling out terminology does not add to the length of the text but improves the experience of the reader. I would encourage the authors to evaluate which abbreviations are truly necessary and eliminate the rest in favor of spelling out those words.

Line 46: “DOC” need to be introduced or (see my comment above) it may be easier to just spell out dissolved organic carbon.

Line 47-48: “close the gap to the chemical analytical data” does not make sense

Line 55-59: Break this up into two sentences at pharmaceuticals.

Line 92: “allows to draw” is missing something

Line 144-145: Swap these two sentences. Everywhere else in the paper upstream is introduced first so continue this first.

Line 145: “Additionally, the WWTP effluent was sampled” this could be added to another sentence.

Line 215-217: Reword these two sentences

Line 223: AChE has already been introduced.

Line 271: add “the” before enzyme

Line 464: WWTP is already introduced

Line 531 and 533: a Reference link is broken - please add the pertinent reference

Line 673: PtEQ was already introduced

Line 674: Again, AChE was already introduced.

Line 676: Change “WW” to wastewater

Line 714 and 721: Micropollutant was already introduced so change to “MP” or avoid the abbreviation altogether.

Line 920: AChE was already introduced

Conclusions: While the conclusions highlight some important methodological lessons from this broad and intensive study, they do actually little to answer the question of whether or not Swiss streams (and similar streams world-wide) are actually adversely impacted by the studies compounds and wastewater effluent. If this comprehensive study of 24 rivers cannot answer this question (and I believe it can), which study will ever be able to answer this question? The authors may want to consider revising the Conclusions to address whether or not the studied compounds in effluent pose a risk to aquatic life in downstream reaches of small to medium size streams.

Fig 1 add sample size to each estrogen (this comments is also relevant to several of the other figures as sample size differs between analyses).

The abundance of individual figures makes it difficult to compare effects across assays. Combining several figures (for example all the bioassays) into one larger figure with multiple graphs would help.

Fig 8: since NP could not be measured in all samples, this graph is a bit misleading as it lists NP as one of the estrogenic compound, but then it does not actually show up in the graph.

Reviewer #2: The present study investigates the ecotoxicological effects caused by WWTP effluent in medium size streams by comparing upstream and downstream sites. It assessed the contribution of pollutants on the observed effects based on chemical analysis and performed a risk assessment based on bioassays results and chemical analysis.

This study presents a very robust dataset where appropriate methods were used for the analysis and it is very well written. I has clear objectives and offers a unique opportunity for the comparison of effects identified by bioassays and chemical analysis. In addition, this study complements several others already published that assessed the same sites. Methods limitations were identified and the contribution on the results is shown and discussed.

There are only few minor points that need further clarification. Along the text a lot of emphasis is given to the assays assessing waterflea reproduction and feeding activity of amphipods, however the results are not presented (only figure included in suppl. material) and briefly included in the discussed. Due to it being highlighted along the text, I was expecting to find it. I believe these should be included, at least as a brief paragraph on the results section.

There are errors on the figure referencing. See lines 505, 531, 533, 551, 565.

The figures attached to the submission are very poor. The text is not legible and overall clarity of them is not good. Please correct.

In summary, I believe that this is a high quality manuscript, presenting a robust dataset that complements others previously published. I am happy to recommend it for publication with minor revision needed.

6. PLOS authors have the option to publish the peer review history of their article (what does this mean?). If published, this will include your full peer review and any attached files.

Reviewer #1: No

Reviewer #2: No

---

## [Author Response · Author response to Decision Letter 0]

18 Nov 2019

Response to reviewers’ comments:

Reviewer #1:

The manuscript by Kienle et al., entitled "Effects of treated wastewater on the ecotoxicity of small streams – unravelling the contribution of chemicals causing effects" describes a far-tanging and in depth study of the effects of treated wastewater effluent on small and mid-size streams in Switzerland. The study is comprehensive, presented with clarity and great detail and provides important comprehensive evidence for the effects of micropollutants released into the aquatic environment through the discharge of treated municipal wastewater effluent. The combination of detailed water chemistry, functional assays and analysis in vitro and in vivo across many streams is noteworthy and adds great strength to the conclusions. The manuscript is extremely well written and only requires minor edits to ready it for publications. Methods are exhaustingly described and referenced and Results are well characterized in the Discussion. Methods and especially the Results are extensively documented in the Supplemental Materials accompanying the main text.

 Thank you very much for this positive feedback. It is highly appreciated.

The manuscript is rather long and would benefit from "light trimming". The second to last paragraph of the Introduction (Lines 98 through 116) is more appropriate for the Discussion (where much of it is repeated already) and its purpose tom place the current study into the context of previous and complementary work could readily be accomplished in adding one sentence to the last paragraph of the discussion.

 Thank you for this suggestion. After careful consideration, we decided to keep the second to last paragraph of the introduction where it is, as its aim is to put the current study into place with previous studies. Then the reader has already an overview on what was done previously. Of course, these studies are again picked up in the discussion, however in different contexts.

A minor issue, but one that effects the readability of the manuscript is the excessive use of abbreviations for terminology that is not used much throughout the manuscript. The authors appeared to have struggled with the abundance of abbreviations as well and frequently switch from abbreviation (for example "MP", AChE, same with TP - which in the US stands for "toilet paper' - not what the authors meant, but somewhat appropriate in the context of the research :-) to the full word throughout the text. Spelling out terminology does not add to the length of the text but improves the experience of the reader. I would encourage the authors to evaluate which abbreviations are truly necessary and eliminate the rest in favor of spelling out those words.

 Thank you for this helpful observation. We checked the use of abbreviations throughout the manuscript and reduced it where possible (no use of MP, TP, E1, E2, EE2, NP, BPA, DOC any more).

Line 46: “DOC” need to be introduced or (see my comment above) it may be easier to just spell out dissolved organic carbon.

 We now spelled out dissolved organic carbon throughout the manuscript.

Line 47-48: “close the gap to the chemical analytical data” does not make sense

 This has now been reformulated as: “… to further explore the missing correlation of bioassay data with chemical analytical data..”

Line 55-59: Break this up into two sentences at pharmaceuticals.

 This has been done.

Line 92: “allows to draw” is missing something

 You are right. Thanks for this hint. We now added “allows to draw conclusions on concentrations and effects of micropollutants in stream ecosystems in stream ecosystems up- and downstream of wastewater treatment plants for…”

Line 144-145: Swap these two sentences. Everywhere else in the paper upstream is introduced first so continue this first.

 This is a good suggestions. We now rewrote the paragraph as follows always mentioning upstream first: “At each study site, one upstream (US) location was chosen as reference site and one downstream (DS) sampling location as impacted site, in addition, the WWTP effluent was sampled. The upstream site was located some 20 m upstream from the effluent discharge. At the downstream sampling site wastewater and river water were completely mixed.”

Line 145: “Additionally, the WWTP effluent was sampled” this could be added to another sentence.

 This has been done (see above).

Line 215-217: Reword these two sentences

 The sentences have been reworded as: “Overall, this method was less robust than the method used to determine the other compounds. An important reason is that most of the data had to be extrapolated between 0 and the lowest calibration standard. Further information on the applied method is provided in SI Table S3.2.

Line 223: AChE has already been introduced.

 Thanks for noticing this. Acetylcholinesterase has now been deleted.

Line 271: add “the” before enzyme

 This has been done.

Line 464: WWTP is already introduced

 You are right. However, as this is a figure caption and as these should be self-explanatory, we aimed at explaining all abbreviations used in the corresponding figure. Therefore, we did not incorporate this comment.

Line 531 and 533: a Reference link is broken - please add the pertinent reference

 Thanks for noticing this. We now have checked all Figure references in the manuscript and added the pertinent reference if the link was broken. 

Line 673: PtEQ was already introduced

 You are right. However, as mentioned above, we aimed at keeping the figure captions self-explanatory and therefore introduced the necessary abbreviations directly there.

Line 674: Again, AChE was already introduced.

 That is true. However, as the abbreviation is not necessary for the figure caption as acetylcholinesterase is just used once, we decided not to use the abbreviation here.

Line 676: Change “WW” to wastewater

 This has been done.

Line 714 and 721: Micropollutant was already introduced so change to “MP” or avoid the abbreviation altogether.

 We decided to avoid the abbreviation MP altogether as it is only used in the introduction.

Line 920: AChE was already introduced

 That’s right. Now only AChE is used.

Conclusions: While the conclusions highlight some important methodological lessons from this broad and intensive study, they do actually little to answer the question of whether or not Swiss streams (and similar streams world-wide) are actually adversely impacted by the studies compounds and wastewater effluent. If this comprehensive study of 24 rivers cannot answer this question (and I believe it can), which study will ever be able to answer this question? The authors may want to consider revising the Conclusions to address whether or not the studied compounds in effluent pose a risk to aquatic life in downstream reaches of small to medium size streams.

 Thank you for this valuable feedback. We now added the following information to the conclusions: “Overall, based on the applied bioassays as well as the measured compounds, photosystem II inhibiting herbicides posed the highest risk for aquatic algae and plants at up- and downstream reaches of the investigated streams. Estrogens played a minor role and conclusions on effects of acetylcholinesterase inhibiting insecticides were difficult due to influences of confounding factors in the applied bioassay. However, several exceedances of mixture risk quotients for organophosphate and carbamate insecticides indicated a potential risk for aquatic organisms, especially invertebrates and vertebrates, at downstream sites. In addition, feeding activity of amphipods and reproduction of water flea might be impaired at downstream reaches due to micropollutants originating from the WWTP.”

Fig 1 add sample size to each estrogen (this comments is also relevant to several of the other figures as sample size differs between analyses).

 Thank you for noticing this. You are right, the exact sample size was not always added to compounds with differing sample sizes. We now checked all the figures and figure captions. Sample size is always provided, either in the figure itself (Fig. 2, 3) or in the figure caption (Fig. 1, 4, 5, 6, 7, 8, 9, 10, 11 and in all figures in the SI). Sample size did differ in Fig. 1, 2, 3, 8, 9, 10 and 11. We now added the sample size to Fig. 1, as this was also done for the other compound groups (Fig. 2 and 3). For the remaining figures we decided to keep them as is. Adding the sample size directly to the figures would make them again more complicated. 

The abundance of individual figures makes it difficult to compare effects across assays. Combining several figures (for example all the bioassays) into one larger figure with multiple graphs would help.

 Thank you for this observation. We can imagine that the number of figures is quite challenging for the reader. We tried to group relevant figures together where possible, marked with big letters, e.g. 6A, B and C. However, after careful consideration, we decided against further combining figures in order to limit the amount of information per figure.

Fig 8: since NP could not be measured in all samples, this graph is a bit misleading as it lists NP as one of the estrogenic compound, but then it does not actually show up in the graph.

 Thank you for this remark. I guess you meant Fig. 9, as this lists the contribution of the individual estrogens. You are right, that NP could only be measured in 12 of the 24 samples. However, the reason why it does not show up in the graph at the second half of sites is rather its low contribution to the overall effects: only 0.3% of the effects could be attributed to this compound. The same is true for BPA which accounted for only 0.4% of the observed estrogenicity. Therefore, we did not change the figure.

Reviewer #2: The present study investigates the ecotoxicological effects caused by WWTP effluent in medium size streams by comparing upstream and downstream sites. It assessed the contribution of pollutants on the observed effects based on chemical analysis and performed a risk assessment based on bioassays results and chemical analysis.

This study presents a very robust dataset where appropriate methods were used for the analysis and it is very well written. I has clear objectives and offers a unique opportunity for the comparison of effects identified by bioassays and chemical analysis. In addition, this study complements several others already published that assessed the same sites. Methods limitations were identified and the contribution on the results is shown and discussed.

 Thank you very much for this positive feedback. It is highly appreciated.

There are only few minor points that need further clarification. Along the text a lot of emphasis is given to the assays assessing waterflea reproduction and feeding activity of amphipods, however the results are not presented (only figure included in suppl. material) and briefly included in the discussed. Due to it being highlighted along the text, I was expecting to find it. I believe these should be included, at least as a brief paragraph on the results section.

 We now added a short paragraph summarizing the results and referring to the figures in the SI.

There are errors on the figure referencing. See lines 505, 531, 533, 551, 565.

 This has also been noted by the second reviewer and now has been corrected.

The figures attached to the submission are very poor. The text is not legible and overall clarity of them is not good. Please correct.

 Thanks for noticing this. We checked the quality of the uploaded figures and adapted them with PANE as advised by the editorial office. The quality of them is appropriate for the journal requirements. However, when compiling the pdf for review, the quality of the figures decreased substantially. In the manuscript this should be fine.

In summary, I believe that this is a high quality manuscript, presenting a robust dataset that complements others previously published. I am happy to recommend it for publication with minor revision needed.

 Thank you very much.

---

## [Editor Report · Decision Letter 1]

25 Nov 2019

Effects of treated wastewater on the ecotoxicity of small streams – unravelling the contribution of chemicals causing effects

PONE-D-19-25149R1

Dear Dr. Kienle,

We are pleased to inform you that your manuscript has been judged scientifically suitable for publication and will be formally accepted for publication once it complies with all outstanding technical requirements.

Congratulations, I look forward to seeing your article published!

With kind regards,

James P. Meador

Section Editor

PLOS ONE

---

## [Editor Report · Acceptance letter]

10 Dec 2019

PONE-D-19-25149R1 

Effects of treated wastewater on the ecotoxicity of small streams – unravelling the contribution of chemicals causing effects 

Dear Dr. Kienle:

I am pleased to inform you that your manuscript has been deemed suitable for publication in PLOS ONE. Congratulations! Your manuscript is now with our production department. 

With kind regards,

on behalf of

Dr. James P. Meador 

Section Editor

PLOS ONE